# Multi-Agent Coordination via Multi-Level Communication

**Ziluo Ding**$^{1,2,*,\ddagger}$ **Zeyuan Liu**$^{1,*}$ **Zhirui Fang**$^{1,*}$ **Kefan Su**$^{2}$ **Liwen Zhu**$^{3}$

**Zongqing Lu**$^{2,\dagger}$

$^{1}$Tsinghua Shenzhen International Graduate School, Tsinghua University,
$^{2}$Peking University, $^{3}$Tencent AI Lab

## Abstract

The partial observability and stochasticity in multi-agent settings can be mitigated by accessing more information about others via communication. However, the coordination problem still exists since agents cannot communicate actual actions with each other at the same time due to the circular dependencies. In this paper, we propose a novel multi-level communication scheme, *Sequential Communication* (SeqComm). SeqComm treats agents asynchronously (the upper-level agents make decisions before the lower-level ones) and has two communication phases. In the negotiation phase, agents determine the priority of decision-making by communicating hidden states of observations and comparing the value of intention, obtained by modeling the environment dynamics. In the launching phase, the upper-level agents take the lead in making decisions and then communicate their actions with the lower-level agents. Theoretically, we prove the policies learned by SeqComm are guaranteed to improve monotonically and converge. Empirically, we show that SeqComm outperforms existing methods in various cooperative multi-agent tasks.

## 1 Introduction

Centralized training with decentralized execution (CTDE) [Lowe et al., 2017] is a popular learning paradigm in cooperative multi-agent reinforcement learning (MARL). Although the centralized value function can be learned to evaluate the joint policy of agents, the decentralized policies of agents are essentially independent. Therefore, a coordination problem arises. That is, agents may make sub-optimal actions by mistakenly assuming others' actions when there exist multiple optimal joint actions [Busoniu et al., 2008]. Communication allows agents to obtain information about others to avoid miscoordination [Jiang et al., 2024]. However, most existing work only focuses on communicating messages, *e.g.,* the information of agents' current observation or historical trajectory [Jiang and Lu, 2018, Singh et al., 2019, Das et al., 2019, Ding et al., 2020]. It is impossible for an agent to acquire other's actions before making decisions since the game model is usually synchronous, *i.e.*, agents make decisions and execute actions simultaneously.

A general approach to solving the coordination problem is to make sure that ties between equally good actions are broken by all agents. One simple mechanism for doing so is to know exactly what others will do and adjust the behavior accordingly under a unique ordering of agents and actions

---

$^{*}$Equal contribution.
$^{\dagger}$Correspondence to Zongqing Lu <zongqing.lu@pku.edu.cn>, Ziluo Ding <ziluoding@baai.ac.cn>
$^{\ddagger}$Work done during an internship at Tsinghua Shenzhen International Graduate School, Tsinghua University.

[Busoniu et al., 2008]. Inspired by this, we reconsider the cooperative game from an asynchronous perspective. In other words, each agent is assigned a priority (*i.e.* order) of decision-making at each step, thus the Stackelberg equilibrium (SE) [Von Stackelberg, 2010] is naturally set up as the learning objective. Specifically, the upper-level agents make decisions before the lower-level agents (Each agent represents a unique level, with upper and lower levels being relative.). Therefore, the lower-level agents can acquire the actual actions of the upper-level agents by communication and make their decisions conditioned on what the upper-level agents would do. Importantly, **we never break the fundamental dynamic,** $p(s_{t+1}|s_t, \boldsymbol{a}^{1:k-1})$**, in the multi-agent system.** The agents make decisions asynchronously but perform actions simultaneously as the default environment setting.

Under this setting, the SE is likely to be Pareto superior to the average Nash equilibrium (NE) in games that require a high cooperation level [Zhang et al., 2020]. However, *is it necessary to decide a specific priority of decision-making for each agent?* Ideally, the optimal joint policy can be decomposed by any orders [Wen et al., 2019], *e.g.,* $\pi^*(a_1, a_2|s) = \pi^*(a_1|s)\pi^*(a_2|s,a_1) = \pi^*(a_2|s)\pi^*(a_1|s,a_2)$. But during the learning process, agents are unlikely to use other agents' optimal actions for gradient calculation, making it still vulnerable to the relative overgeneralization problem [Wei et al., 2018]. This means there is no guarantee that different orders will converge to the same suboptimal. We also claim that the different priorities of decision-making may affect the optimality of the convergence of the learning algorithm in Section 3. Note that relative overgeneralization occurs when a suboptimal NE in the joint space of actions is preferred over an optimal NE because each agent's action in the suboptimal equilibrium is a better choice when matched with arbitrary actions from the cooperative agents.

This work proposes a novel multi-level communication scheme for cooperative MARL, *Sequential Communication* (SeqComm), to enable agents to coordinate with each other explicitly. Specifically, SeqComm has two-phase communication, negotiation phase and launching phase. In the negotiation phase, agents communicate their hidden states of observations with others simultaneously. Then, they can generate multiple predicted trajectories, called *intention*, by modeling the environmental dynamics and other agents' actions. In addition, the priority of decision-making is determined by communicating and comparing the agents' intentions, which are evaluated by their state-value functions. **The value of each intention represents the predicted rewards obtained by treating that agent as the first mover of the order sequence.** The sequence of others follows the same procedure as aforementioned with the upper-level agents fixed. In the launching phase, the upper-level agents take the lead in decision-making and communicate their actual actions with the lower-level agents. The actual actions will be executed simultaneously in the environment without changes.

SeqComm is currently built on MAPPO [Yu et al., 2021]. Theoretically, we prove the policies learned by SeqComm are guaranteed to improve monotonically and converge. Empirically, we evaluate SeqComm on StarCraft multi-agent challenge v2 (SMACv2) [Samvelyan et al., 2019]. We demonstrate that SeqComm outperforms existing communication-free and communication-based methods in various maps in SMACv2. By ablation studies, we confirm that treating agents asynchronously is a more effective way to promote coordination, and SeqComm can provide the proper priority of decision-making for agents to develop better coordination.

## 2 Related Work

**Communication.** Existing work [Jiang and Lu, 2018, Kim et al., 2019, Singh et al., 2019, Das et al., 2019, Zhang et al., 2019, Jiang et al., 2020, Ding et al., 2020, Konan et al., 2022] in this realm mainly focus on how to extract valuable messages. ATOC [Jiang and Lu, 2018] and IC3Net [Singh et al., 2019] utilize gate mechanisms to decide when to communicate with other agents. Several studies [Das et al., 2019, Konan et al., 2022] employ multi-round communication to fully reason the intentions of others and establish complex collaboration strategies. Social influence [Jaques et al., 2019] uses communication to influence the behaviors of others. I2C [Ding et al., 2020] only communicates with agents that are relevant and influential which are determined by causal inference. However, all these methods focus on how to exploit valuable information from current or past partial observations effectively and properly. More recently, some studies [Kim et al., 2021, Du et al., 2021, Pretorius et al., 2021] begin to answer the question: can we favor cooperation beyond sharing partial observation? They allow agents to imagine their future states with a world model and communicate those with others. IS [Pretorius et al., 2021], as the representation of this line of research, enables each agent to share its intention with other agents in the form of the encoded imagined trajectory and use the

attention module to figure out the importance of the received intention. However, two concerns arise. On one hand, circular dependencies can lead to inaccurate predicted future trajectories as long as the multi-agent system treats agents synchronously. On the other hand, MARL struggles in extracting useful information from numerous messages, not to mention more complex and dubious messages, *i.e.* predicted future trajectories. Unlike these studies, we treat the agents from an asynchronous perspective, therefore, circular dependencies can be naturally resolved. Moreover, agents send actions to lower-level agents, making the messages compact and informative.

**Coordination.** The agents are essentially independent decision-makers in execution and may break ties between equally good actions randomly. Thus, in the absence of additional mechanisms, different agents may break ties in different ways, and the resulting joint actions may be suboptimal. Coordination graphs [Guestrin et al., 2002, Böhmer et al., 2020, Wang et al., 2021] simplify the coordination when the global Q-function can be additively decomposed into local Q-functions that only depend on the actions of a subset of agents. Typically, a coordination graph expresses a higher-order value decomposition among agents. This improves the representational capacity to distinguish other agents' effects on local utility functions, which addresses the miscoordination problems caused by partial observability.

Another general approach to solving the coordination problem is to make sure that ties are broken by all agents in the same way, requiring that random action choices are somehow coordinated or negotiated. Social conventions [Boutilier, 1996] or role assignments [Prasad et al., 1998] encode prior preferences towards certain joint actions and help break ties during action selection. Communication [Fischer et al., 2004, Vlassis, 2007] can be used to negotiate action choices, either alone or in combination with the aforementioned techniques. Our method follows this line of research by utilizing the ordering of agents and actions to break the ties, other than the enhanced representational capacity of the local value function.

More discussions of related work are in Appendix C.

## 3 Problem Formulation

**Cost-Free Communication.** The decentralized partially observable Markov decision process (Dec-POMDP) can be extended to multi-agent POMDP [Oliehoek et al., 2016] by sharing observations among agents via communication. The joint observations are not necessarily equivalent to the state. However, joint observations can be used to represent the state better than single observations.

Previous work [Pynadath and Tambe, 2002] shows that under cost-free communication, agents would share optimal messages for mutual interest. If the communication cost is high, there is a balance between delivering all the useful messages for greater benefits and keeping the amount of communication as low as possible. In addition, analyzing this extreme case gives us some understanding of the benefit of communication, even if the results do not apply across all domains. However, even under multi-agent POMDP where agents can get joint observations, coordination problems can still arise [Busoniu et al., 2008]. Suppose the centralized critic has learnt actions pairs $[a_1, a_2]$ and $[b_1, b_2]$ that are equally optimal. Without any prior information, the individual policies $\pi_1$ and $\pi_2$ learned from the centralized critic can break the ties randomly and may choose $a_1$ and $b_2$, respectively.

**Multi-Agent Sequential Decision-Making.** We consider fully cooperative multi-agent tasks that are modeled as multi-agent POMDP, where $n$ agents interact with the environment according to the following procedure, which we refer to as *multi-agent sequential decision-making*.

At each timestep $t$, assume the priority (*i.e.* order) of decision-making for all agents is given and each priority level has only one agent (*i.e.*, agents make decisions one by one). Note that the smaller the level index, the higher priority of decision-making is. The agent at each level $k$ gets its own observation $o_t^k$ drawn from the state $s_t$, and receives messages $\boldsymbol{m}_t^{-k}$ from all other agents, where $\boldsymbol{m}_t^{-k} \triangleq \{\{o_t^1, a_t^1\}, \ldots, \{o_t^{k-1}, a_t^{k-1}\}, o_t^{k+1}, \ldots, o_t^n\}$. Equivalently, $\boldsymbol{m}_t^{-k}$ can be written as $\{\boldsymbol{o_t}^{-k}, \boldsymbol{a}_t^{1:k-1}\}$, where $\boldsymbol{o_t}^{-k}$ denotes the joint observations of all agents except $k$ (in practice, agents communicate the hidden states/encodings of observations), and $\boldsymbol{a}_t^{1:k-1}$ denotes the joint actions of agents 1 to $k-1$. For the agent at the first level (*i.e.*, $k = 1$), $\boldsymbol{a}_t^{1:k-1} = \varnothing$. Then, the agent determines its action $a_t^k$ sampled from its policy $\pi_k(\cdot|o_t^k, \boldsymbol{m}_t^{-k})$ or equivalently $\pi_k(\cdot|\boldsymbol{o}_t, \boldsymbol{a}_t^{1:k-1})$ and sends it to the lower-level agents. After all, agents have determined their actions, they perform the joint actions

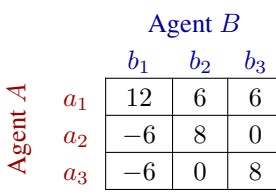
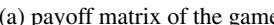

|        | Agent $B$ | | |
|--------|-----------|------|------|
|        | $b_1$ | $b_2$ | $b_3$ |
| $a_1$  | 12 | 6 | 6 |
| $a_2$  | $-6$ | 8 | 0 |
| $a_3$  | $-6$ | 0 | 8 |

Agent $A$

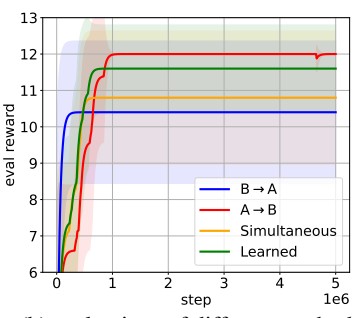

(a) payoff matrix of the game      (b) evaluations of different methods

Figure 1: (a) Payoff matrix for a one-step game. There are multiple local optima. (b) Evaluations of different methods for the game in terms of the mean reward and standard deviation of ten runs. $A \rightarrow B$, $B \rightarrow A$, *Simultaneous*, and *Learned* represent that agent $A$ makes decisions first, agent $B$ makes decisions first, two agents make decisions simultaneously, and there is another learned policy determining the priority of decision making, respectively. MAPPO [Yu et al., 2021] is used as the backbone.

$a_t$, which can be seen as sampled from the joint policy $\boldsymbol{\pi}(\cdot|s_t)$ *factorized* as $\prod_{k=1}^{n} \pi_k(\cdot|\boldsymbol{o}_t, \boldsymbol{a}_t^{1:k-1})$, in the environment and get a shared reward $r(s_t, \boldsymbol{a}_t)$ and the state transitions to next state $s'$ according to the transition probability $p(s'|s_t, \boldsymbol{a}_t)$. All agents aim to maximize the expected return $\sum_{t=0}^{\infty} \gamma^t r_t$, where $\gamma$ is the discount factor. The state-value function and action-value function of the level-$k$ agent are defined as follows:

$$V_{\pi_k}(s, \boldsymbol{a}^{1:k-1}) \triangleq \mathop{\mathbb{E}}_{\substack{s_{1:\infty} \\ \boldsymbol{a}_0^{k:n} \sim \boldsymbol{\pi}_{k:n} \\ \boldsymbol{a}_{1:\infty} \sim \boldsymbol{\pi}}} \left[ \sum_{t=0}^{\infty} \gamma^t r_t | s_0 = s, \boldsymbol{a}_0^{1:k-1} = \boldsymbol{a}^{1:k-1} \right]$$

$$Q_{\pi_k}(s, \boldsymbol{a}^{1:k}) \triangleq \mathop{\mathbb{E}}_{\substack{s_{1:\infty} \\ \boldsymbol{a}_0^{k+1:n} \sim \boldsymbol{\pi}_{k+1:n} \\ \boldsymbol{a}_{1:\infty} \sim \boldsymbol{\pi}}} \left[ \sum_{t=0}^{\infty} \gamma^t r_t | s_0 = s, \boldsymbol{a}_0^{1:k} = \boldsymbol{a}^{1:k} \right].$$

For the setting of multi-agent sequential decision-making discussed above, we have the following proposition.

**Proposition 1.** *If all the agents update their policy with individual TRPO [Schulman et al., 2015] sequentially in multi-agent sequential decision-making, then the joint policy of all agents are guaranteed to improve monotonically and converge.*

*Proof.* The proof is given in Appendix A. □

Proposition 1 indicates that SeqComm has the performance guarantee regardless of the priority of decision-making in multi-agent sequential decision-making. However, the priority of decision-making indeed affects the optimality of the converged joint policy, and we have the following claim.

**Claim 1.** *The different priorities of decision-making affect the optimality of the convergence of the learning algorithm due to the relative overgeneralization problem.*

We use a one-step matrix game as an example, as illustrated in Figure 1(a), to demonstrate the influence of the priority of decision-making on the learning process. Due to relative overgeneralization [Wei et al., 2018], agent $B$ tends to choose $b_2$ or $b_3$. Specifically, $b_2$ or $b_3$ in the suboptimal equilibrium is a better choice than $b_1$ in the optimal equilibrium when matched with arbitrary actions from agent $A$. Therefore, as shown in Figure 1(b), $B \rightarrow A$ (*i.e.*, agent $B$ makes decisions before $A$, and $A$'s policy conditions on the action of $B$) and *Simultaneous* (*i.e.*, two agents make decisions simultaneously and independently) are easily trapped into local optima. However, if agent $A$ goes first, things can be different, as $A \rightarrow B$ achieves the optimum. As long as agent $A$ does not suffer from relative overgeneralization, it can help agent $B$ get rid of local optima by narrowing down the search space of $B$. Besides, a policy that determines the priority of decision-making can be learned under the

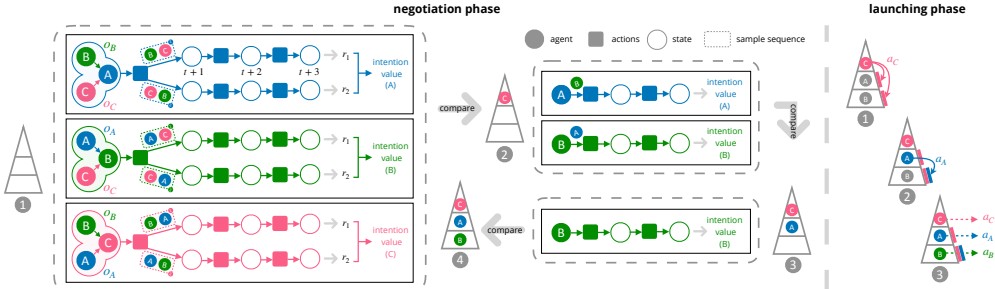

Figure 2: Overview of SeqComm. SeqComm has two communication phases, the negotiation phase (*left*) and the launching phase (*right*). In the negotiation phase, agents communicate hidden states of observations with others and obtain their own intention. The priority of decision-making is determined by sharing and comparing the value of all the intentions. In the launching phase, the agents who hold the upper-level positions will make decisions prior to the lower-level agents. Besides, their actions will be shared with anyone that has not yet made decisions.

guidance of the state-value function, denoted as *Learned*. It obtains better performance than $B \to A$ and *Simultaneous*, which indicates that dynamically determining the order during policy learning can be beneficial as we do not know the optimal priority in advance.

**Remark 1.** The priority (*i.e.* order) of decision-making affects the optimality of the converged joint policy in multi-agent sequential decision-making, thus it is critical to determine the order. However, learning the order directly requires an additional centralized policy in execution, which is not generalizable in a scenario where the number of agents varies. Moreover, its learning complexity exponentially increases with the number of agents, making it infeasible in many cases.

## 4 Sequential Communication

In this paper, we cast our eyes in another direction and resort to the world model, which is the dynamic model of the environment. Ideally, we can randomly sample candidate order sequences, evaluate them under the world model (see Section 4.1), and choose the order sequence that is deemed the most promising under the true dynamic. SeqComm is designed based on this principle to determine the priority of decision-making via communication.

In SeqComm, communication is separated into phases serving different purposes and multi-round communication in one phase is possible. One is the *negotiation* phase for agents to determine the priority of decision-making. Another is the *launching* phase for agents to act conditioning on actual actions upper-level agents will take to implement *explicit coordination via communication*. The overview of SeqComm is illustrated in Figure 2. Each SeqComm agent consists of a policy, a critic, and a world model, as illustrated in Figure 3, and the parameters of all networks are shared across agents [Gupta et al., 2017].

**World Model.** The world model is needed to predict and evaluate future trajectories. SeqComm, unlike previous works [Kim et al., 2021, Du et al., 2021, Pretorius et al., 2021], can utilize received hidden states of other agents in the first round of communication to model more precise environment dynamics for the explicit coordination in the next round of communication. Once an agent can access other agents' hidden states, it shall have adequate information to estimate their actions since all agents are homogeneous and parameter-sharing. Therefore, the world model $\mathcal{M}(\cdot)$ takes as input the joint hidden states $\boldsymbol{h}_t = \{h_t^1, \ldots, h_t^n\}$ and actions $\boldsymbol{a}_t$, and predicts the next joint observations and reward. In practice, before the inputs pass into the world model, the attention module $\mathrm{AM_w}$ is utilized to process the input.

$$\hat{\boldsymbol{o}}_{t+1}, \hat{r}_{t+1} = \mathcal{M}_i(\mathrm{AM_w}(\boldsymbol{h}_t, \boldsymbol{a}_t)).$$

The reason that we adopt the attention module is to entitle the world model to be generalizable in the scenarios where additional agents are introduced or existing agents are removed.

## 4.1 Negotiation Phase

In the negotiation phase, the observation encoder first takes $o_t$ as input and outputs a hidden state $h_t$ to compress the information, which is used to communicate with others. Note that many studies [Ding et al., 2020, Jiang and Lu, 2018] found that redundant messages may impair the learning process empirically. In more detail, the model can converge slowly or sometimes lead to a worse sub-optimal. Agents then determine the priority of decision-making by *intentions* which is established and evaluated based on the world model.

**Priority of Decision-Making.** Intention is the key element in determining the priority of decision-making. The notion of intention is described as an agent's future behavior in previous works [Rabinowitz et al., 2018, Raileanu et al., 2018, Kim et al., 2021]. However, we define the *intention* as an agent's future behavior *without considering others*.

As mentioned before, an agent's intention considering others can lead to circular dependencies and cause miscoordination. By our definition, the intention of an agent should be depicted as all future trajectories considering that agent as the first mover and ignoring the others. However, there are many possible future trajectories as the priority of the rest of the agents is *unfixed*. In practice, we use the Monte Carlo method to estimate the intention value based on all future trajectories. Note that it is uniform across priorities for unfixed agents. Each order should be treated equally since we do not have any prior for the distribution.

Taking agent $i$ at timestep $t$ to illustrate, it firstly considers itself as the first-mover and produces its action only based on the joint hidden states, $\hat{a}_t^i \sim \pi_i(\cdot|\mathrm{AM_a}(\boldsymbol{h}_t, \emptyset))$, where we again use an attention module $\mathrm{AM_a}$ to handle the input. For the order sequence of lower-level agents, we randomly sample a set of order sequences from unfixed agents. Assume agent $j$ is the second-mover, agent $i$ models $j$'s action by considering the upper-level action following its own policy $\hat{a}_t^j \sim \pi_i(\cdot|\mathrm{AM_a}(\boldsymbol{h}_t, \hat{a}_t^i))$. The same procedure is applied to predict the actions of all other agents following the sampled order sequence. Based on the joint hidden states and predicted actions, the next joint observations $\hat{\boldsymbol{o}}_{t+1}$ can be predicted by the world model $\mathcal{M}$. The length of the predicted future trajectory is $H$ and it can then be written as $\tau^t = \{\hat{\boldsymbol{o}}_{t+1}, \hat{\boldsymbol{a}}_{t+1}, \ldots, \hat{\boldsymbol{o}}_{t+H}, \hat{\boldsymbol{a}}_{t+H}\}$ by repeating the procedure aforementioned.

Then, the agent uses its critic (state-value function) to evaluate the future trajectory and output value $v_{\tau^t}$. The intention value is defined as the average value of $F$ future trajectories with different sampled order sequences. *Through the critic, we have linked the order and agent performance together.*

After all the agents have computed their intentions and the corresponding value, they again communicate their intention values to others. Then, agents would compare and choose the agent with the highest intention value to be the first mover. The priority of lower-level decision-making follows the same procedure with the upper-level agents fixed. Note that some agents may communicate intention values with others multiple times until the priority of decision-making is finally determined.

## 4.2 Negotiation Phase for Local Communication

The full communication version of SeqComm is constructed based on theoretical derivation. It has a theoretical guarantee to some extent, but some assumptions, e.g., broadcast communication, can be unrealistic and incur lots of communication overhead. Therefore, we provide another version of SeqComm in scenarios where agents can only communicate with nearby agents (agents within a limited communication range).

In more detail, the agent first calculates its intention value based only on the hidden states of nearby agents. After comparing with the intention values of nearby agents (intention values are communicated with the nearby agents), the agent can determine the upper-level and lower-level nearby agents. Unlike the previous version of SeqComm, agents cannot distinguish the detailed order sequence of the upper-level nearby agents since their communication ranges may not overlap. Therefore, the intention values are calculated and communicated among agents for *only one time*. The local communication version greatly reduces communication overhead, making it more suitable for many real applications.

For more details of the algorithms, please refer to the Appendix D for the pseudo-code.

## 4.3 Launching Phase

As for the launching phase, agents communicate to obtain additional information to make decisions. Apart from the received hidden states from the last phase, we allow agents to get what *actual* actions the upper-level agents will take in execution, while other studies can only infer others' actions by opponent modeling [Rabinowitz et al., 2018, Raileanu et al., 2018] or communicating intentions [Kim et al., 2021]. Therefore, miscoordination can be naturally avoided, and a better cooperation strategy is possible since lower-level agents can adjust their behaviors accordingly.

A lower-level agent $i$ make a decision following the policy $\pi_i(\cdot|\text{AM}_a(\boldsymbol{h}_t, \boldsymbol{a}_t^{upper}))$, where $\boldsymbol{a}_t^{upper}$ means received actual actions from all upper-level agents. As long as the agent has decided on an action, it will send the action to all other lower-level agents through the communication channel. *Note that the actions are executed simultaneously and distributedly in execution, though agents make decisions sequentially.*

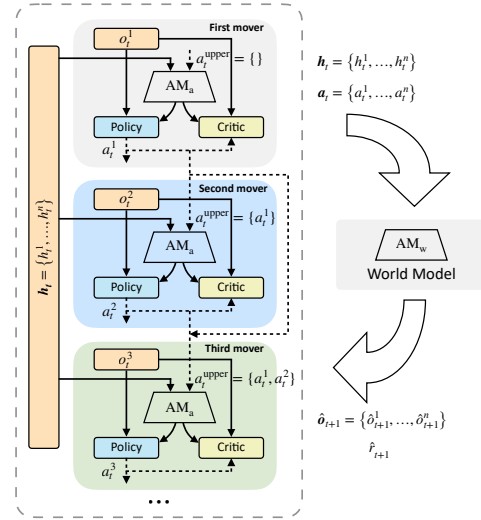

Figure 3: Architecture of SeqComm. The critic and policy of each agent take input as its own observation and received messages. The world model takes as input the joint hidden states and predicted joint actions.

### 4.4 Theoretical Analysis

As intention values determine the priority of decision-making, SeqComm is likely to choose different orders *at different timesteps* during training. However, we have the following proposition that theoretically guarantees the performance of the learned joint policy under SeqComm.

**Proposition 2.** *The monotonic improvement and convergence of the joint policy in SeqComm are independent of the priority of decision-making of agents at each timestep.*

*Proof.* The proof is given in Appendix A. □

The priority of decision-making is chosen under the world model, thus the compounding errors in the world model can result in discrepancies between the predicted returns of the same order under the world model and the true dynamics. We then analyze the monotonic improvement for the joint policy under the world model based on Janner et al. [2019].

**Theorem 1.** *Let the expected total variation between two transition distributions be bounded at each timestep as $\max_t \mathbb{E}_{s \sim \boldsymbol{\pi}_{\beta,t}}[D_{TV}(p(s'|s,\boldsymbol{a})||\hat{p}(s'|s,\boldsymbol{a}))] \leq \epsilon_m$, and the policy divergences at level $k$ be bounded as $\max_{s,\boldsymbol{a}^{1:k-1}} D_{TV}(\pi_{\beta,k}(a^k|s,\boldsymbol{a}^{1:k-1})||\pi_k(a^k|s,\boldsymbol{a}^{1:k-1})) \leq \epsilon_{\pi_k}$, where $\boldsymbol{\pi}_\beta$ is the data collecting policy for the model and $\hat{p}(s'|s,\boldsymbol{a})$ is the transition distribution under the model. Then the model return $\hat{\eta}$ and true return $\eta$ of the policy $\boldsymbol{\pi}$ are bounded as:*

$$\hat{\eta}[\boldsymbol{\pi}] \geq \eta[\boldsymbol{\pi}] - \underbrace{[\frac{2\gamma r_{\max}(\epsilon_m + 2\sum_{k=1}^n \epsilon_{\pi_k})}{(1-\gamma)^2} + \frac{4r_{\max}\sum_{k=1}^n \epsilon_{\pi_k}}{(1-\gamma)}]}_{C(\epsilon_m, \boldsymbol{\epsilon}_{\pi_{1:n}})}.$$

*Proof.* The proof is given in Appendix B. □

**Remark 2.** Theorem 1 provides a useful relationship between the compounding errors and the policy update. As long as we improve the return under the true dynamic by more than the gap, $C(\epsilon_m, \boldsymbol{\epsilon}_{\pi_{1:n}})$, we can guarantee the policy improvement under the world model. If no such policy exists to overcome the gap, it implies the model error is too high, that is, there is a large discrepancy between the world model and true dynamics. Thus the order sequence obtained under the world model is not reliable. Such an order sequence is almost the same as a random one. Though a random order sequence also has the theoretical guarantee of Proposition 2, we will show in Section 5.2 that a random order sequence leads to a poor local optimum empirically.

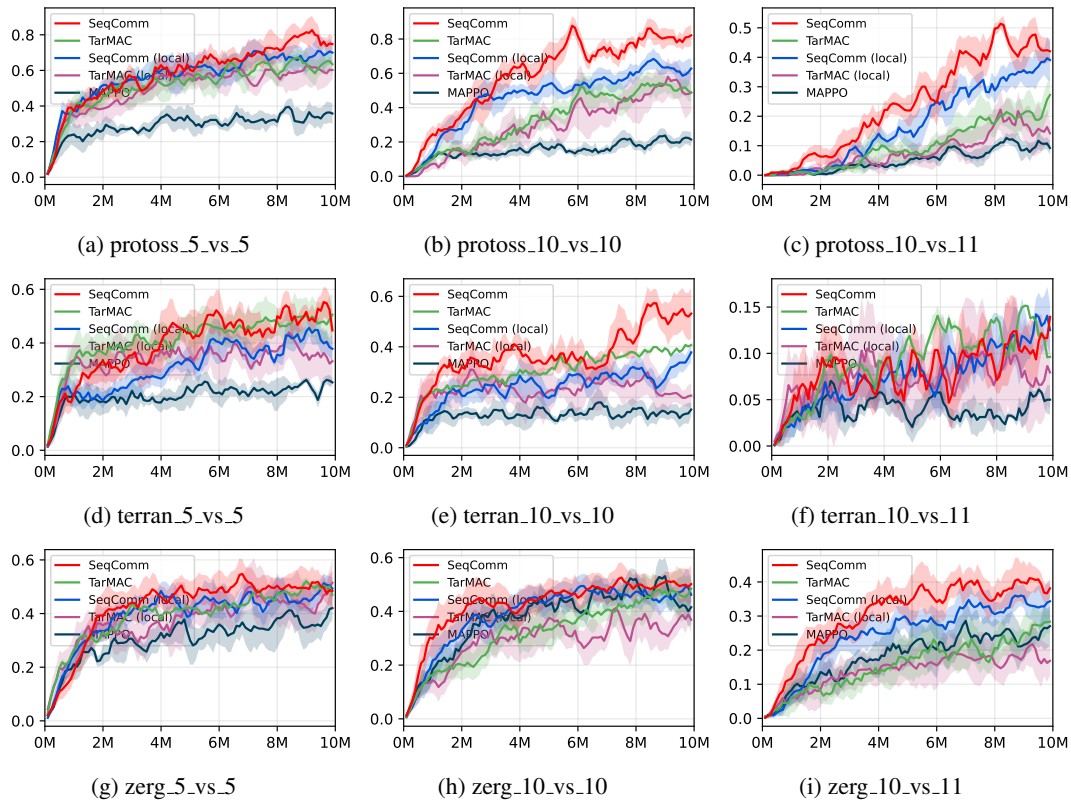

Figure 4: Learning curves of SeqComm and baselines in nine SMACv2 maps.

# 5 Experiments

SeqComm is currently instantiated based on MAPPO [Yu et al., 2021]. We evaluate SeqComm nine maps in StarCraft multi-agent challenge v2 (SMACv2) [Ellis et al., 2024].

In the experiments, SeqComm and baselines are parameter-sharing for fast convergence [Gupta et al., 2017, Terry et al., 2020]. We have fine-tuned the baselines for a fair comparison. *The world model in the SMACv2 environment is trained from scratch and kept fine-tuned in the learning process.* Therefore, no extra prior knowledge is provided. Please refer to the Appendix for the hyperparameter settings. All results are presented in terms of the mean and standard deviation of five runs with different random seeds.

## 5.1 Results

**SMACv2.** We have evaluated our method on the *most representative and challenging* multi-agent environment currently available. Compared with SMAC [Samvelyan et al., 2019], SMACv2 has some better properties, *i.e.* stochasticity and partial observability. In other words, agents need to cooperate more in the new environment to complete tasks, whereas they could achieve a certain success rate without cooperation in the original environment.

We have chosen nine maps for extensive evaluation and made some minor changes to the observation part of agents to make it more difficult. Specifically, the sight range of agents is reduced from 9 to 3, and agents cannot perceive any information about their allies even if they are within the sight range. NDQ [Wang et al., 2020] adopts a similar change to increase the difficulty of action coordination. The rest of the settings remain the same as the default. In summary, *we require the environment to be one where a high success rate cannot be achieved solely based on individual observations.*

We also evaluate the local communication version of SeqComm. Agents can only communicate with nearby agents (agents within their communication range). Note that the map size and the total number

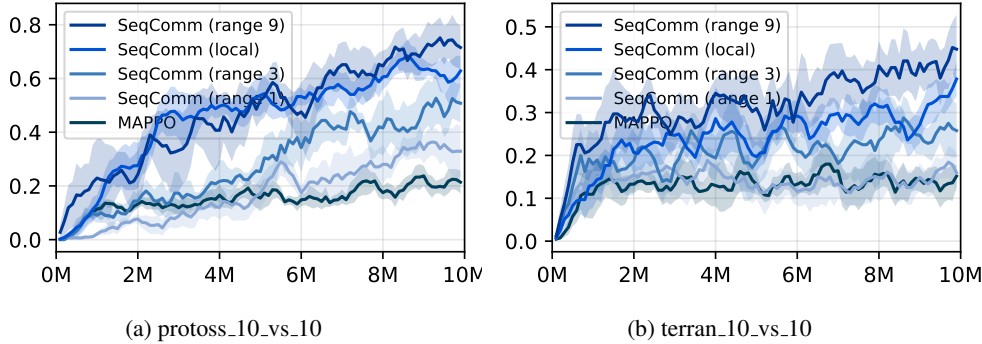

(a) protoss_10_vs_10              (b) terran_10_vs_10

Figure 5: Ablation studies of the communication ranges.

of agents restrict the number of nearby agents. As the task progresses, the number of nearby agents is from 2 to 4.

**Analysis.** The learning curves of SeqComm and the baselines in terms of the win rate are illustrated in Figure 4. All communication-based methods perform better than communicaion-free method (MAPPO). In easy scenarios, communication may not be very useful, but experiments have shown that in cases with significant partial observability and stochasticity, communication can greatly enhance agent ability.

We compare our method with TarMAC [Das et al., 2019], which holds a similar position in communication settings to that of MAPPO in communication-free settings. SeqComm outperforms TarMAC in all maps, which verifies the gain of explicit action coordination. Moreover, the full-communication version performs better than the local-communication version because the former can access more information. However, it also costs more communication overhead.

## 5.2 Ablation Studies

**Priority of Decision-Making.** We primarily want to contribute a practical version to the community. Moreover, *the fewer communicative agents there are, the fewer possible orders there are, thus increasing the probability of randomly obtaining a good order*. It would be more meaningful to demonstrate that devoting effort to finding a good order is still important in such a scenario. Therefore, we do the ablation study for the local version of the SeqComm. In more detail, we compare SeqComm with two ablation baselines: the priority of decision-making is determined randomly at each timestep, denoted as *Random*, and agents only access the observations of others during training and execution, denoted as *No action*.

As depicted in Figure 7, SeqComm achieves a higher win rate than *Random* and *No action* in all the maps. These results verify the importance of the priority of decision-making and the necessity to adjust it continuously during one episode. It is also demonstrated that SeqComm can provide a proper priority of decision-making. As discussed in Section 4.4, although *Random* also has the theoretical guarantee, they converge to poor local optima in practice. Surprisingly, in most tasks, *Random* performs worse than *No action*. It again verifies that a bad order may fail to improve coordination or even impair it.

**Communication Range.** We also conduct experiments to demonstrate the impact of different communication ranges. We set communication ranges to {1, 3, 9}, in addition to the default range of 6. We notice a steady improvement in performance as the communication range increases. Therefore, the choice of communication range is a trade-off between communication overhead and agent performance. In our previous experiments, we choose a compromise value of 3 for the local version to validate the effectiveness of our method. Results refer to Figure 5.

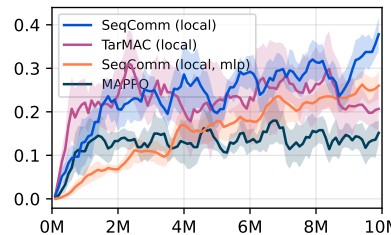

Figure 6: Ablation studies on the network mechanisms.

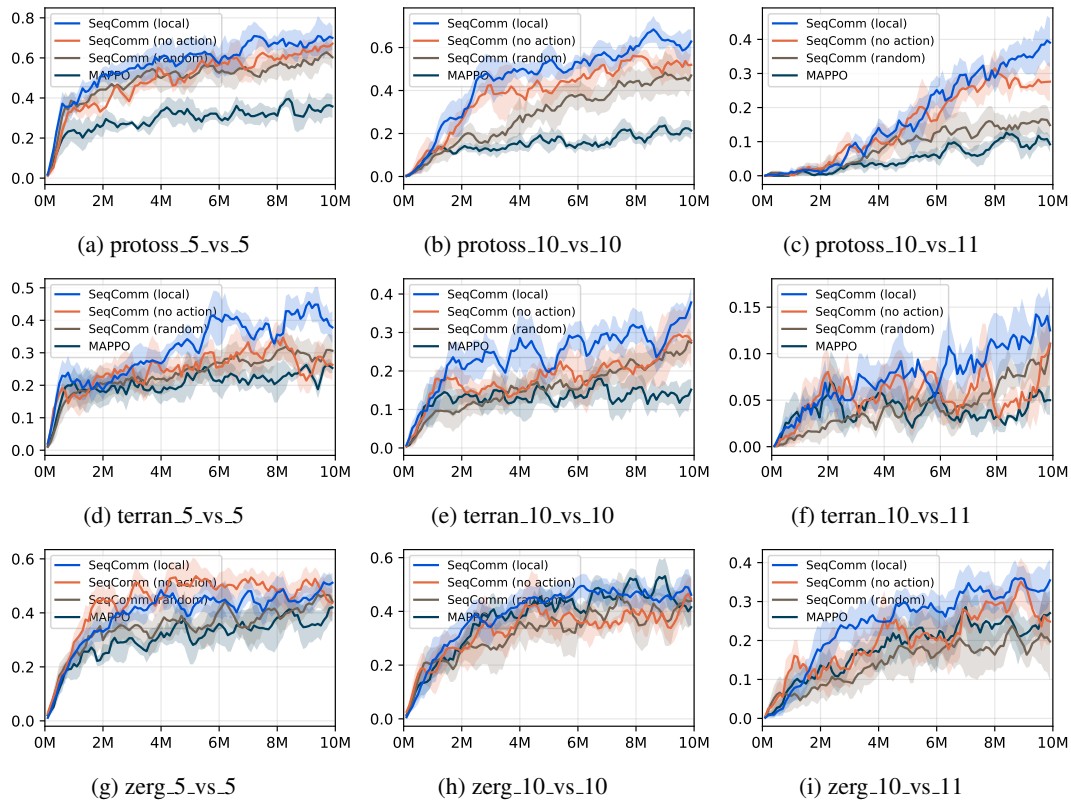

Figure 7: Ablation studies under local communication in SMACv2.

**Network Mechanisum.** We replaced the attention mechanism for local communication with an aggregation method. In more detail, messages are concatenated and passsd into a five-layer linear neural networks. The curve is based on 3 random seeds and tested on the terran 10v10 map. The results refer to Figure 6.

# 6  Conclusions

We have proposed SeqComm, which enables agents to coordinate well and explicitly with each other, and it, from an asynchronous perspective, allows agents to make decisions sequentially. A two-phase communication scheme has been adopted to determine the priority of decision-making and transfer messages accordingly. Empirically, it is demonstrated that SeqComm outperforms baselines in a variety of cooperative multi-agent scenarios.

**Limitations.** The assumption of accessing the local observation of any other agent could be strong since it is unsuitable for all applications. Thus, we provide a local communication version of SeqComm for assumption relaxation in the experiment.

# Acknowledgements

This work was supported by the STI 2030-Major Projects under Grant 2021ZD0201404 and the NSFC under Grants 62450001 and 62476008.

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

# A    Proofs of Proposition 1 and Proposition 2

**Lemma 1** (Agent-by-Agent PPO). *If we update the policy of each agent $i$ with TRPO Schulman et al. [2015] (or approximately PPO) when fixing all the other agent's policies, then the joint policy will improve monotonically.*

*Proof.* We consider the joint surrogate objective in TRPO $L_{\boldsymbol{\pi}_{\mathrm{old}}}(\boldsymbol{\pi}_{\mathrm{new}})$ where $\boldsymbol{\pi}_{\mathrm{old}}$ is the joint policy before updating and $\boldsymbol{\pi}_{\mathrm{new}}$ is the joint policy after updating.

Given that $\pi_{\mathrm{new}}^{-i} = \pi_{\mathrm{old}}^{-i}$, we have:

$$
\begin{aligned}
L_{\boldsymbol{\pi}_{\mathrm{old}}}(\boldsymbol{\pi}_{\mathrm{new}}) &= \mathbb{E}_{a \sim \boldsymbol{\pi}_{\mathrm{new}}}[A_{\boldsymbol{\pi}_{\mathrm{old}}}(s, \boldsymbol{a})] \\
&= \mathbb{E}_{a \sim \boldsymbol{\pi}_{\mathrm{old}}}\Big[\frac{\boldsymbol{\pi}_{\mathrm{new}}(\boldsymbol{a}|s)}{\boldsymbol{\pi}_{\mathrm{old}}(\boldsymbol{a}|s)} A_{\boldsymbol{\pi}_{\mathrm{old}}}(s, \boldsymbol{a})\Big] \\
&= \mathbb{E}_{a \sim \boldsymbol{\pi}_{\mathrm{old}}}\Big[\frac{\pi_{\mathrm{new}}^i(a^i|s)}{\pi_{\mathrm{old}}^i(a^i|s)} A_{\boldsymbol{\pi}_{\mathrm{old}}}(s, \boldsymbol{a})\Big] \\
&= \mathbb{E}_{a^i \sim \pi_{\mathrm{old}}^i}\Big[\frac{\pi_{\mathrm{new}}^i(a^i|s)}{\pi_{\mathrm{old}}^i(a^i|s)} \mathbb{E}_{a^{-i} \sim \pi_{old}^{-i}}[A_{\boldsymbol{\pi}_{\mathrm{old}}}(s, a^i, a^{-i})]\Big] \\
&= \mathbb{E}_{a^i \sim \pi_{\mathrm{old}}^i}\Big[\frac{\pi_{\mathrm{new}}^i(a^i|s)}{\pi_{\mathrm{old}}^i(a^i|s)} A_{\boldsymbol{\pi}_{\mathrm{old}}}^i(s, a^i)\Big] \\
&= L_{\pi_{\mathrm{old}}^i}(\pi_{\mathrm{new}}^i),
\end{aligned}
$$

where $A_{\boldsymbol{\pi}_{\mathrm{old}}}^i(s, a^i) = \mathbb{E}_{a^{-i} \sim \pi_{\mathrm{old}}^{-i}}[A_{\boldsymbol{\pi}_{\mathrm{old}}}(s, a^i, a^{-i})]$ is the individual advantage of agent $i$, and the third equation is from the condition $\pi_{\mathrm{new}}^{-i} = \pi_{\mathrm{old}}^{-i}$.

With the result of TRPO, we have the following conclusion:

$$
\begin{aligned}
J(\pi_{\mathrm{new}}) - J(\pi_{\mathrm{old}}) &\geq L_{\boldsymbol{\pi}_{\mathrm{old}}}(\boldsymbol{\pi}_{\mathrm{new}}) - C D_{\mathrm{KL}}^{\max}(\boldsymbol{\pi}_{\mathrm{new}}||\boldsymbol{\pi}_{\mathrm{old}}) \\
&= L_{\pi_{\mathrm{old}}^i}(\pi_{\mathrm{new}}^i) - C D_{\mathrm{KL}}^{\max}(\pi_{\mathrm{new}}^i||\pi_{\mathrm{old}}^i) \quad \text{(from } \pi_{\mathrm{new}}^{-i} = \pi_{\mathrm{old}}^{-i})
\end{aligned}
$$

This means the individual objective is the same as the joint objective so the monotonic improvement is guaranteed. $\qquad\square$

Then we can show the proof of Proposition 1.

*Proof.* We will build a new MDP $\tilde{M}$ based on the original MDP. We keep the action space $\tilde{A} = A = \times_{i=1}^n A^i$, where $A^i$ is the original action space of agent $i$. The new state space contains multiple layers. We define $\tilde{S}^k = S \times (\times_{i=1}^k A^i)$ for $k = 1, 2, \cdots, n-1$ and $\tilde{S}^0 = S$, where $S$ is the original state space. Then a new state $\tilde{s}^k \in \tilde{S}^k$ means that $\tilde{s}^k = (s, a^1, a^2, \cdots, a^k)$. The total new state space is defined as $\tilde{S} = \cup_{i=0}^{n-1} \tilde{S}^i$. Next we define the transition probability $\tilde{P}$ as following:

$$
\begin{aligned}
\tilde{P}(\tilde{s}'|\tilde{s}^k, a^{k+1}, a^{-(k+1)}) &= \mathbb{1}\left(\tilde{s}' = (\tilde{s}^k, a^{k+1})\right), \ k < n-1 \\
\tilde{P}(\tilde{s}'|\tilde{s}^k, a^{k+1}, a^{-(k+1)}) &= \mathbb{1}\left(\tilde{s}' \in \tilde{S}^0\right) P(\tilde{s}'|\tilde{s}^k, a^{k+1}), \ k = n-1.
\end{aligned}
$$

This means that the state in the layer $k$ can only transition to the state in the layer $k + 1$ with the corresponding action, and the state in the layer $n - 1$ will transition to the layer 0 with the probability $P$ in the original MDP. The reward function $\tilde{r}$ is defined as following:

$$
\tilde{r}(\tilde{s}, \boldsymbol{a}) = \mathbb{1}\left(\tilde{s} \in \tilde{S}_0\right) r(\tilde{s}, \boldsymbol{a}).
$$

This means the reward is only obtained when the state in layer 0 and the value is the same as the original reward function. Now we obtain the total definition of the new MDP $\tilde{M} = \{\tilde{S}, \tilde{A}, \tilde{P}, \tilde{r}, \gamma\}$.

Then we claim that if all agents learn in multi-agent sequential decision-making by PPO, they are actually taking agent-by-agent PPO in the new MDP $\tilde{M}$. To be precise, one update of multi-agent

sequential decision-making in the original MDP $M$ equals to a round of update from agent 1 to agent $n$ by agent-by-agent PPO in the new MDP $\tilde{M}$. Moreover, the total reward of a round in the new MDP $\tilde{M}$ is the same as the reward in one timestep in the original MDP $M$. With this conclusion and Lemma 1, we complete the proof.

$\square$

The proof of Proposition 2 can be seen as a corollary of the proof of Proposition 1.

*Proof.* From Lemma 1 we know that the monotonic improvement of the joint policy in the new MDP $\tilde{M}$ is guaranteed for each update of one single agent's policy. So even if the different round of updates in the new MDP $\tilde{M}$ is with different order of the decision-making, the monotonic improvement of the joint policy is still guaranteed. Finally, from the proof of Proposition 1, we know that the monotonic improvement in the new MDP $\tilde{M}$ equals to the monotonic improvement in the original MDP $M$. These complete the proof. $\square$

## B  Proofs of Theorem 1

**Lemma 2** (TVD of the joint distributions). *Suppose we have two distribution $p_1(x, y) = p_1(x)p_1(x|y)$ and $p_2(x, y) = p_2(x)p_2(x|y)$. We can bound the total variation distance of the joint as:*

$$D_{TV}(p_1(x,y)||p_2(x,y)) \leq D_{TV}(p_1(x)||p_2(x)) + \max_x D_{TV}(p_1(y|x)||p_2(y|x))$$

*Proof.* See [Janner et al., 2019] (Lemma B.1). $\square$

**Lemma 3** (Markov chain TVD bound, time-varing). *Suppose the expected KL-divergence between two transition is bounded as $\max_t \mathbb{E}_{s \sim p_{1,t}(s)} D_{KL}(p_1(s'|s)||p_2(s'|s)) \leq \delta$, and the initial state distributions are the same $p_{1,t=0}(s) = p_{2,t=0}(s)$. Then the distance in the state marginal is bounded as:*

$$D_{TV}(p_{1,t}(s)||p_{2,t}(s)) \leq t\delta$$

*Proof.* See [Janner et al., 2019] (Lemma B.2). $\square$

**Lemma 4** (Branched Returns Bound). *Suppose the expected KL-divergence between two dynamics distributions is bounded as $\max_t \mathbb{E}_{s \sim p_{1,t}(s)}[D_{TV}(p_1(s'|s, \boldsymbol{a})||p_2(s'|s, \boldsymbol{a}))]$, and the policy divergences at level $k$ are bounded as $\max_{s, \boldsymbol{a}^{1:k-1}} D_{TV}(\pi_1(a^k|s, \boldsymbol{a}^{1:k-1})||\pi_2(a^k|s, \boldsymbol{a}^{1:k-1})) \leq \epsilon_{\pi_k}$. Then the returns are bounded as:*

$$|\eta_1 - \eta_2| \leq \frac{2r_{\max}\gamma(\epsilon_m + \sum_{k=1}^n \epsilon_{\pi_k})}{(1-\gamma)^2} + \frac{2r_{\max}\sum_{k=1}^n \epsilon_{\pi_k}}{1-\gamma},$$

*where $r_{\max}$ is the upper bound of the reward function.*

*Proof.* Here, $\eta_1$ denotes the returns of $\boldsymbol{\pi}_1$ under dynamics $p_1(s'|s, \boldsymbol{a})$, and $\eta_2$ denotes the returns of $\boldsymbol{\pi}_2$ under dynamics $p_2(s'|s, \boldsymbol{a})$. Then we have

$$
\begin{aligned}
|\eta_1 - \eta_2| &= |\sum_{s, \boldsymbol{a}} (p_1(s, \boldsymbol{a}) - p_2(s, \boldsymbol{a}))r(s, \boldsymbol{a})| \\
&= |\sum_t \sum_{s, \boldsymbol{a}} \gamma^t (p_{1,t}(s, \boldsymbol{a}) - p_{2,t}(s, \boldsymbol{a}))r(s, \boldsymbol{a})| \\
&\leq \sum_t \sum_{s, \boldsymbol{a}} \gamma^t |p_{1,t}(s, \boldsymbol{a}) - p_{2,t}(s, \boldsymbol{a})|r(s, \boldsymbol{a}) \\
&\leq r_{\max} \sum_t \sum_{s, \boldsymbol{a}} \gamma^t |p_{1,t}(s, \boldsymbol{a}) - p_{2,t}(s, \boldsymbol{a})|.
\end{aligned}
$$

By Lemma 2, we get

$$
\begin{aligned}
\max_s D_{TV}(\pi_1(\boldsymbol{a}|s)||\pi_2(\boldsymbol{a}|s)) \leq{}& \max_{s,a_1} D_{TV}(\pi_1(\boldsymbol{a}^{-1}|s,a^1)||\pi_2(\boldsymbol{a}^{-1}|s,a^1)) \\
& + \max_s D_{TV}(\pi_1(a^1|s)||\pi_2(a^1|s)) \\
\leq{}& \cdots \\
\leq{}& \sum_{k=1}^{n} \max_{s,\boldsymbol{a}^{1:k-1}} D_{TV}(\pi_1(a^k|s,\boldsymbol{a}^{1:k-1})||\pi_2(a^k|s,\boldsymbol{a}^{1:k-1})) \\
\leq{}& \sum_{k=1}^{n} \epsilon_{\pi_k}.
\end{aligned}
$$

We then apply Lemma 3, using $\delta = \epsilon_m + \sum_{k=1}^{n} \epsilon_{\pi_k}$ (via Lemma 3 and 2) to get

$$
\begin{aligned}
D_{TV}(p_{1,t}(s)||p_{2,t}(s)) \leq{}& t \max_t E_{s \sim p_{1,t}(s)} D_{TV}(p_{1,t}(s'|s)||p_{2,t}(s'|s)) \\
\leq{}& t \max_t E_{s \sim p_{1,t}(s)} D_{TV}(p_{1,t}(s',\boldsymbol{a}|s)||p_{2,t}(s',\boldsymbol{a}|s)) \\
\leq{}& t(\max_t E_{s \sim p_{1,t}(s)} D_{TV}(p_{1,t}(s'|s,\boldsymbol{a})||p_{2,t}(s'|s,\boldsymbol{a})) \\
& + \max_t E_{s \sim p_{1,t}(s)} \max_s D_{TV}(\boldsymbol{\pi}_{1,t}(\boldsymbol{a}|s)||\boldsymbol{\pi}_{2,t}(\boldsymbol{a}|s))) \\
\leq{}& t(\epsilon_m + \sum_{k=1}^{n} \epsilon_{\pi_k})
\end{aligned}
$$

And we also get $D_{TV}(p_{1,t}(s,\boldsymbol{a})||p_{2,t}(s,\boldsymbol{a})) \leq t(\epsilon_m + \sum_{k=1}^{n} \epsilon_{\pi_k}) + \sum_{k=1}^{n} \epsilon_{\pi_k}$ by Lemma 2. Thus, by plugging this back, we get:

$$
\begin{aligned}
|\eta_1 - \eta_2| \leq{}& r_{\max} \sum_t \sum_{s,\boldsymbol{a}} \gamma^t |p_{1,t}(s,\boldsymbol{a}) - p_{2,t}(s,\boldsymbol{a})| \\
\leq{}& 2r_{\max} \sum_t \gamma^t (t(\epsilon_m + \sum_{k=1}^{n} \epsilon_{\pi_k}) + \sum_{k=1}^{n} \epsilon_{\pi_k}) \\
\leq{}& 2r_{\max} \left( \frac{\gamma(\epsilon_m + \sum_{k=1}^{n} \epsilon_{\pi_k})}{(1-\gamma)^2} + \frac{\sum_{k=1}^{n} \epsilon_{\pi_k}}{1-\gamma} \right)
\end{aligned}
$$

$\square$

Then we can show the proof of Theorem 1.

*Proof.* Let $\boldsymbol{\pi}_\beta$ denote the data collecting policy. We use Lemma 4 to bound the returns, but it will require bounded model error under the new policy $\boldsymbol{\pi}$. Thus, we need to introduce $\boldsymbol{\pi}_\beta$ by adding and subtracting $\eta[\boldsymbol{\pi}_\beta]$, to get:

$$
\hat{\eta}[\boldsymbol{\pi}] - \eta[\boldsymbol{\pi}] = \hat{\eta}[\boldsymbol{\pi}] - \eta[\boldsymbol{\pi}_\beta] + \eta[\boldsymbol{\pi}_\beta] - \eta[\boldsymbol{\pi}].
$$

we can bound $L_1$ and $L_2$ both using Lemma 4 by using $\delta = \sum_{k=1}^{n} \epsilon_{\pi_k}$ and $\delta = \epsilon_m + \sum_{k=1}^{n} \epsilon_{\pi_k}$ respectively, and obtain:

$$
L_1 \geq -\frac{2\gamma r_{\max} \sum_{k=1}^{n} \epsilon_{\pi_k}}{(1-\gamma)^2} - \frac{2r_{\max} \sum_{k=1}^{n} \epsilon_{\pi_k}}{(1-\gamma)}
$$

$$
L_2 \geq -\frac{2\gamma r_{\max}(\epsilon_{\pi_m} + \sum_{k=1}^{n} \epsilon_{\pi_k})}{(1-\gamma)^2} - \frac{2r_{\max} \sum_{k=1}^{n} \epsilon_{\pi_k}}{(1-\gamma)}.
$$

Adding these two bounds together yields the conclusion. $\square$

# C   Additional Related Work

**Reinforcement Learning in Stackelberg Game**   Many previous studies [Könönen, 2004, Sodomka et al., 2013, Greenwald et al., 2003, Zhang et al., 2020] have investigated reinforcement learning in finding Stackelberg equilibrium. Bi-AC [Zhang et al., 2020] is a bi-level actor-critic method that allows agents to have different knowledge base so that Stackelberg equilibrium (SE) is possible to find. The actions can still be executed simultaneously and distributedly. It empirically studies the relationship between the cooperation level and the superiority of Stackelberg equilibrium to Nash equilibrium. AQL [Könönen, 2004] updates the Q-value by solving the SE in each iteration and can be regarded as the value-based version of Bi-AC.

Existing work mainly focuses on two-agent settings, and their order is fixed in advance. However, fixed order can hardly be an optimal solution, especially for large-scale homogeneous agent scenarios. To address this issue, we exploit agents' intentions to dynamically determine the priority of decision-making along the way of interacting with each other.

**Multi-Agent Path Finding (MAPF)**   MAPF aims to plan collision-free paths for multiple agents on a given graph from their given start vertices to target vertices. In MAPF, prioritized planning is deeply coupled with collision avoidance [Van Den Berg and Overmars, 2005, Ma et al., 2019], where collision is used to design constraints or heuristics for planning.

We will distinguish MAPF from our work from three perspectives, *i.e.* problem definition, the motivation behind agent ordering, and the incompatibility of the two methods.

Problem definition: MAPF aims to plan collision-free paths for multiple agents on a given graph from their given start vertices to their given target vertices. However, we aim to find a communication-based solution for any Markov decision process with interests aligned. MDP covers lots of possible coordination-needed scenarios, not just avoiding collisions. Besides, each agent has no specific given target.

Motivation: In MAPF, prioritized planning does not offer completeness or optimality guarantees. It is nevertheless popular because of its efficiency. In addition, the order is mainly used for avoiding collision. Unlike MAPF, our main contribution is to introduce prioritized decision-making to MARL and a method to determine the priority of decision-making. To the best of our knowledge, determining the priority of decision-making for learning algorithms has not been investigated. Moreover, combining with learning algorithms will make prioritized decision-making more general (solving MDPs), not just motion planning.

Methods: The different motivations and problems to solve will lead to the incompatibility of the algorithms in the two fields. For MAPF, the order is assigned arbitrarily or derived from the problem at hand. Collision is the keyword and prioritized planning is deeply coupled with this specific coordination problem so that better performance can be achieved. Taking the method Ma et al. [2019] as an example, their two algorithms are conflict-driven search frameworks. That is, collision is used to design some constraints which are guided for search. In MARL, we have lots of unseen coordination problems and we cannot enumerate them all to design constraints.

# D   Implementation Details

## D.1   Algorithm

In this part, we provide the pseudo-code of SeqComm as below:

---
**Algorithm 1** Negotiation Phase

---
**Require:** Number of agents $N$
    $\mathcal{P} = [\,]$: already determined priority
    $\mathcal{A} = \{1, 2, ..., N\}$: remaining agents
    /* Agents communicate the hidden state $\boldsymbol{h}$ of their observations with each other*/
    **for** $i = 1, 2, ..., N$ **do**
        **for** $j$ **in** $\mathcal{A}$ **do**
            Compute agent $j$'s intention value $v_j$ via Algorithm 2
        **end for**

---

/* Agents in $\mathcal{A}$ communicate the intention values with each other*/
Set $p_i$ to be the agent $j$ with the maximum $v_j$
Append $p_i$ to $\mathcal{P}$ and remove it from $\mathcal{A}$
**end for**

---

**Algorithm 2** Intention Value Calculation of Agent $a$

---

**Require:** Already determined priority $\mathcal{P}$, remaining agents $\mathcal{A}$, number of sampling trajectories $F$, length of predicted future trajectory $H$, policy $\pi$ and attention module $\text{AM}_\text{a}$, world model $\mathcal{M}$ and attention module $\text{AM}_\text{w}$, discount factor $\gamma$
  **for** $i = 1, 2, ..., F$ **do**
    Randomly shuffle $\mathcal{A} \setminus \{a\}$ to sample a decision-making priority $\mathcal{P}_{\mathcal{A} \setminus \{a\}}$ of the remaining agents except agent $a$
    **for** $j = 0, 1, ..., H - 1$ **do**
      $\hat{\boldsymbol{a}}^{upper} = \{\}$: predicted actions from all upper-level agents
      **for** $k$ **in** $\text{Concat}(\mathcal{P}, [a], \mathcal{P}_{\mathcal{A} \setminus \{a\}})$ **do**
        Sample $\hat{a}^k$ following $\pi(\cdot | \text{AM}_\text{a}(\boldsymbol{h}_{t+j}, \boldsymbol{a}^{upper}))$
        Append $\hat{a}^k$ to $\hat{\boldsymbol{a}}^{upper}$
      **end for**
      Rollout one step with the world model $\hat{\boldsymbol{o}}_{t+j+1}, \hat{r}_{t+j+1} = \mathcal{M}(\text{AM}_\text{w}(\boldsymbol{h}_{t+j}, \boldsymbol{a}^{upper}))$
    **end for**
    Compute the return of the trajectory $v_i = \sum_{t'=t+1}^{t+H} \gamma^{t'-t-1} \hat{r}_{t'}$ via the critic
  **end for**
  Compute the average return $v = \frac{1}{F} \sum_{i=1}^{F} v_i$

---

**Algorithm 3** Launching Phase

---

**Require:** Decision-making priority $\mathcal{P}$, policy $\pi$ and $\text{AM}_\text{a}$
  $\boldsymbol{a}_t^{upper} = \{\}$: actions from all upper-level agents
  **for** $i$ **in** $\mathcal{P}$ **do**
    Sample $a_t^i$ following $\pi_i(\cdot | \text{AM}_\text{a}(\boldsymbol{h}_t, \boldsymbol{a}_t^{upper}))$
    Append $a_t^i$ to $\boldsymbol{a}_t^{upper}$
    /* Send $\boldsymbol{a}_t^{upper}$ to the lower agent*/
  **end for**
  Interact with the environment with $\boldsymbol{a}_t$

---

We also provide the pseudo-code of the local communication version as below:

---

**Algorithm 4** Local Negotiation Phase of Agent $a$

---

**Require:** Neighbouring agents $\mathcal{N}$
  /* Agents communicate the hidden state $\boldsymbol{h}$ of their observations with neighbouring agents*/
  Compute local intention $v_a$ via Algorithm 5
  /* Send $v_a$ to neighbouring agents and receive $\{v_i\}_{i \in \mathcal{N}}$ from them */
  Set upper-level neighbouring agents $\mathcal{N}^{upper} = \{i \mid v_i > v_a, i \in \mathcal{N}\}$
  Set lower-level neighbouring agents $\mathcal{N}^{lower} = \{i \mid v_i < v_a, i \in \mathcal{N}\}$

---

**Algorithm 5** Local Intention Value Calculation of Agent $a$

---

**Require:** Neighbouring agents $\mathcal{N}$, number of sampling trajectories $F$, length of predicted future trajectory $H$, policy $\pi$ and $\text{AM}_\text{a}$, world model $\mathcal{M}$ and $\text{AM}_\text{w}$, discount factor $\gamma$
  **for** $i = 1, 2, ..., F$ **do**
    Randomly shuffle $\mathcal{N}$ to sample a local decision-making priority $\mathcal{P}_{\mathcal{N}}$
    **for** $j = 0, 1, ..., H - 1$ **do**
      $\hat{\boldsymbol{a}}^{upper} = \{\}$: predicted actions from all upper-level agents
      **for** $k$ **in** $\text{Concat}([a], \mathcal{P}_{\mathcal{N}})$ **do**
        Sample $\hat{a}^k$ following $\pi(\cdot | \text{AM}_\text{a}(\boldsymbol{h}_{t+j}, \boldsymbol{a}^{upper}))$
        Append $\hat{a}^k$ to $\hat{\boldsymbol{a}}^{upper}$
      **end for**
      Rollout one step with the world model $\hat{\boldsymbol{o}}_{t+j+1}, \hat{r}_{t+j+1} = \mathcal{M}(\text{AM}_\text{w}(\boldsymbol{h}_{t+j}, \boldsymbol{a}^{upper}))$

**end for**
    Compute the return of the trajectory $v_i = \sum_{t'=t+1}^{t+H} \gamma^{t'-t-1} \hat{r}_{t'}$
**end for**
Compute the average return $v = \frac{1}{F} \sum_{i=1}^{F} v_i$ via the critic

---

**Algorithm 6** Local Launching Phase of Agent $a$

---

**Require:** Upper-level neighbouring agents $\mathcal{N}^{upper}$, lower-level neighbouring agents $\mathcal{N}^{lower}$, policy
    $\pi$ and $\text{AM}_a$
    /* Receive upper-level actions $\boldsymbol{a}_t^{upper}$ from all upper-level neighbouring agents $\mathcal{N}^{upper}$ /*
    Sample $a_t^i$ following $\pi_i(\cdot | \text{AM}_a(\boldsymbol{h}_t, \boldsymbol{a}_t^{upper}))$
    /* Send $a_t^i$ to all lower-level neighbouring agents $\mathcal{N}^{lower}$ */
    Interact with the environment with $\boldsymbol{a}_t$

---

### D.2 Architecture and Hyperparameters

Our models, including SeqComm and its ablations, are implemented based on MAPPO. Two fully connected layers realize the critic and policy network. As for the attention module, key, query, and value have one fully connected layer each. The size of the hidden layers is 100. Tanh functions are used as nonlinearity. As there is no released code of TarMAC, we implement TarMAC by ourselves, following the instructions mentioned in the original papers [Das et al., 2019].

For the world model, observations and actions are firstly encoded by a fully connected layer. The output size for the observation encoder is 48, and the output size for the action encoder is 16. Then, the outputs of the encoder will be passed into the attention module using the same structure aforementioned. Finally, we use a fully connected layer to decode. In these layers, Tanh is used as the nonlinearity.

SeqComm and its ablation baseline share the same hyperparameters. For Protoss, the learning rate is $1e^{-5}$, while for Terran and Zerg, the learning rate is $2.5e^{-5}$. $H$ and $F$ for calculating intention value is set to 20 and 2. For TarMAC, the learning rate is tuned as $5e^{-5}$. TarMAC adopts MAPPO as the backbone and two-round communication mechanism. For MAPPO, we follow the default settings of the official code [Yu et al., 2021].

### D.3 Attention Module

Attention module (AM) is applied to process messages in the world model, critic network, and policy network. AM consists of three components: query, key, and values. The output of AM is the weighted sum of values, where the weight of value is determined by the dot product of the query and the corresponding key.

For AM in the world model denoted as $\text{AM}_w$, agent $i$ gets messages $\boldsymbol{m}_t^{-i} = \boldsymbol{h}_t^{-i}$ from all other agents at timestep $t$ in negotiation phase, and predicts a query vector $q_t^i$ following $\text{AM}_{w,q}^i(h_t^i)$. The query is used to compute a dot product with keys $\boldsymbol{k}_t = [k_t^1, \cdots, k_t^n]$. Note that $k_t^j$ is obtained by the message from agent $j$ following $\text{AM}_{a,k}^i(h_t^j)$ for $j \neq i$, and $k_t^i$ is from $\text{AM}_{neg,k}^i(h_t^i)$. Besides, it is scaled by $1/\sqrt{d_k}$ followed by a softmax to obtain attention weights $\alpha$ for each value vector:

$$\alpha_i = \text{softmax} \left[ \frac{q_t^{iT} k_t^1}{\sqrt{d_k}} \cdots \underbrace{\frac{q_t^{iT} k_t^j}{\sqrt{d_k}}}_{\alpha_{ij}} \cdots \frac{q_t^{iT} k_t^n}{\sqrt{d_k}} \right] \tag{1}$$

The output of attention module is defined as: $c_t^i = \sum_{j=1}^{n} \alpha_{ij} v_t^j$, where $v_t^j$ is obtained from messages or its own hidden state of observation following $\text{AM}_{w,v}^i(\cdot)$.

As for AM in the policy and critic network denoted as $\text{AM}_a$, agent $i$ gets additional messages from upper-level agent in the launching phase. The message from upper-level and lower-level agent can be expanded as $\boldsymbol{m}_t^{upper} = [\boldsymbol{h}_t^{upper}, \boldsymbol{a}_t^{upper}]$ and $\boldsymbol{m}_t^{lower} = [\boldsymbol{h}_t^{lower}, 0]$, respectively. In addition, the query depends on agent's own hidden state of observation $h_t^i$, but keys and values are only from messages of other agents.

### D.4 Training

The training of SeqComm is an extension of MAPPO. The observation encoder $e$, the critic $V$, and the policy $\pi$ are respectively parameterized by $\theta_e$, $\theta_v$, $\theta_\pi$. Besides, the attention module $\mathrm{AM_a}$ is parameterized by $\theta_a$ and takes as input the agent's hidden state, the messages (hidden states of other agents) in the negotiation phase, and the messages (the actions of upper-level agents) in launching phase. Let $\mathcal{D} = \{\tau_k\}_{k=1}^K$ be a set of trajectories by running policy in the environment. Note that we drop time $t$ in the following notations for simplicity.

The value function is fitted by regression on mean-squared error:

$$\mathcal{L}(\theta_v, \theta_a, \theta_e) = \frac{1}{KT} \sum_{\tau \in \mathcal{D}} \sum_{t=0}^{T-1} \left\| V(\mathrm{AM_a}(e(\boldsymbol{o}), \boldsymbol{a}^{upper})) - \hat{R} \right\|_2^2 \tag{2}$$

where $\hat{R}$ is the discount rewards-to-go.

We update the policy by maximizing the PPO-Clip objective:

$$\mathcal{L}(\theta_\pi, \theta_a, \theta_e) = \frac{1}{KT} \sum_{\tau \in \mathcal{D}} \sum_{t=0}^{T-1} \min\left(\frac{\pi(a|\mathrm{AM_a}(e(\boldsymbol{o}), \boldsymbol{a}^{upper}))}{\pi_{old}(a|\mathrm{AM_a}(e(\boldsymbol{o}), \boldsymbol{a}^{upper}))} A_{\pi_{old}}, g(\epsilon, A_{\pi_{old}})\right) \tag{3}$$

where $g(\epsilon, A) = \begin{cases} (1+\epsilon)A & A \geq 0 \\ (1-\epsilon)A & A \leq 0 \end{cases}$, and $A_{\pi_{old}}(\boldsymbol{o}, \boldsymbol{a}^{upper}, a)$ is computed using the GAE method.

The world model $\mathcal{M}$ is parameterized by $\theta_w$ is trained as a regression model using the training data set $\mathcal{S}$. It is updated with the loss:

$$\mathcal{L}(\theta_w) = \frac{1}{|\mathcal{S}|} \sum_{\boldsymbol{o}, \boldsymbol{a}, \boldsymbol{o}', r \in \mathcal{S}} \left\| (\boldsymbol{o}', r) - \mathcal{M}(\mathrm{AM_w}(e(\boldsymbol{o}), \boldsymbol{a})) \right\|_2^2. \tag{4}$$

We trained our model on one GeForce GTX 1050 Ti and Intel(R) Core(TM) i9-9900K CPU @ 3.60GHz.

### D.5 Addtional Ablation Studies

We conduct a comparison of SeqComm against MAIC and CommFormer across six different maps: Protoss, Terran, and Zerg in 5v5 scenarios (first row) and Protoss, Terran, and Zerg in 10v10 scenarios (second row). The evaluation uses the official codebase for each method, with three random seeds per map under a full communication setting. The results refer to Figure 8.

### D.6 Emergence of Behavioral Patterns

We have visualized several key frames in Figure 9 to highlight the observed behavioral patterns. In the combat game, concentrating attacks on a single enemy is consistently more effective than dispersing them. In frames 1-3, the agents lack specific targets until one agent, located at the end of the orange arrow, approaches an enemy in the bottom right corner. By frame 4, following the negotiation phase, this agent is designated as the highest-level agent (level 5), given its advantageous position to choose an enemy to attack. Once lower-level agents receive the actions from higher-level agents (represented by the white dashed line), all the red units cease random roaming and instead coordinate a unified attack on the blue units. A similar pattern can be observed in frames 7-9.

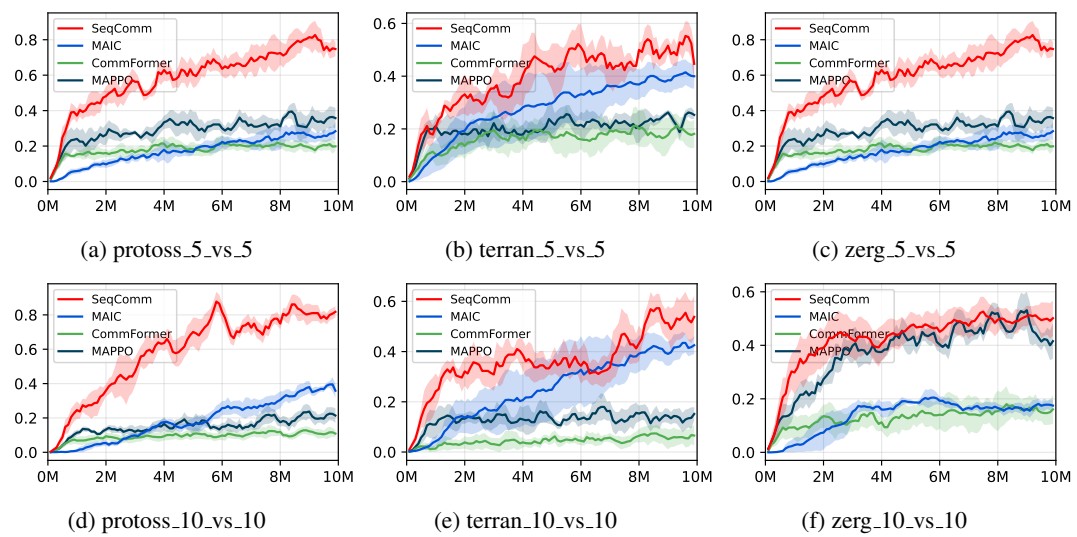

(a) protoss_5_vs_5    (b) terran_5_vs_5    (c) zerg_5_vs_5

(d) protoss_10_vs_10    (e) terran_10_vs_10    (f) zerg_10_vs_10

Figure 8: Results of experiments with extra baseline algorithms.

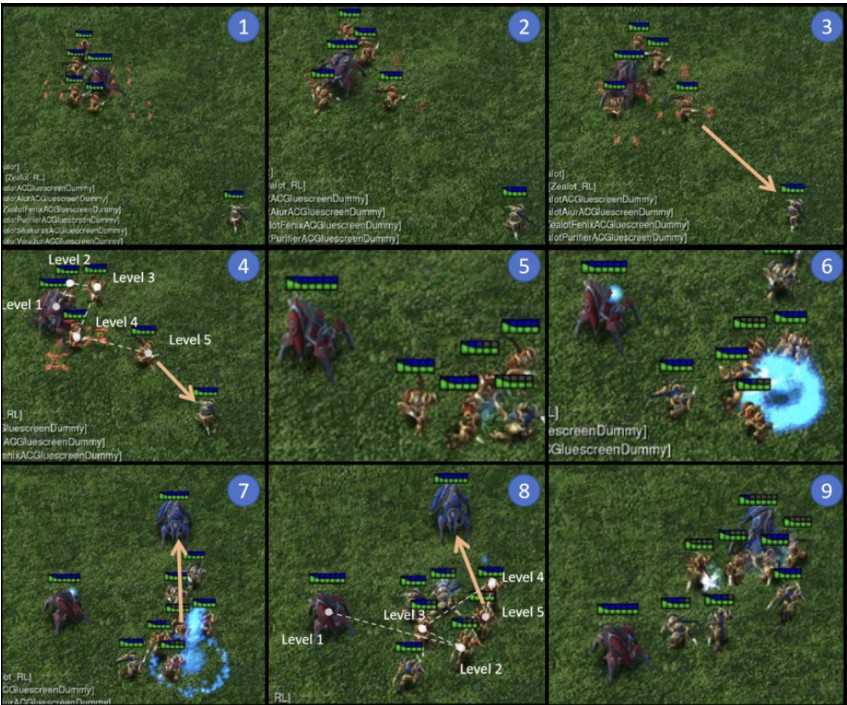

Figure 9: Illustration of the Emergence of Behavioral Patterns

