# OpenReview forum: "Multi-Agent Coordination via Multi-Level Communication"
_NeurIPS.cc/2024/Conference — NeurIPS 2024 poster_

### Official Review · Reviewer_gF4i · 2024-06-27

**Soundness:** 2
**Presentation:** 3
**Contribution:** 2
**Rating:** 5
**Confidence:** 4

**Summary:**

This paper proposed a multi-level sequential communication framework that has two communication phases. It also proved that the policies learned in this way are guaranteed to improve and converge. Instead of the observations, this paper focus on make agents communicate about the action selections. Each agent is assigned a priority of decision-making first and an equilibrium is set up to be the learning objective for all agents working together.

**Strengths:**

This paper is well-written and well motivated, it borrows ideas from game theory and try to bring the equilibrium into multi-agent decision-making process.

**Weaknesses:**

1. If the authors formulate this problem as multi-agent sequential decision making problem, the Markovian property required for formulating the problem as an MDP is no longer satisfied and there is a need proving an alternative policy gradient theorem before simply inserting the Seqcomm into MAPPO.
2. In Figure 2's caption, the authors stated that 'in the launching phase, the agents who hold the upper-level positions will make decisions prior to the lower-level agents. Besides, their actions will be shared with anyone that has not yet made decisions', then how to change the structure of MAPPO model to make it fit this change? And how to adjust the time step based on this change?
3. There is no code available in this submission, the details of the models' structures could not be seen. The reproducibility remains unsure.
4. Only comparing to TarMAC is not enough, this MARL with communication algorithm was proposed in 2019, there are more recent MARL communication algorithms outperforming TarMAC.

**Questions:**

See the weakness section above.

**Limitations:**

Yes.

---

> ### Author Rebuttal · Authors · 2024-08-07
>
> To begin with, we thank the reviewer for the carefully reviewing and insightful advice.
>
> >If the authors formulate this problem as a multi-agent sequential decision-making problem, the Markovian property required for formulating the problem as an MDP is no longer satisfied and there is a need to prove an alternative policy gradient theorem before simply inserting the Seqcomm into MAPPO.
>
> To connect SeqComm and MAPPO while avoiding this problem, we build a new MDP $\tilde{M}$ based on the original MDP $M$. The state space of $\tilde{M}$ can be divided into multiple layers and the state transitions only occur between consecutive layers. The sequential decision-making process in the original MDP corresponds to a round of agent-by-agent decision-making process while the Markovian property of $\tilde{M}$ is preserved. We show SeqComm in original MDP equals Agent-by-agent PPO in new MDP $\tilde{M}$. More detailed discussions are included in Appendix A.
>
> >In Figure 2's caption, the authors stated that 'in the launching phase, the agents who hold the upper-level positions will make decisions prior to the lower-level agents. Besides, their actions will be shared with anyone that has not yet made decisions, then how to change the structure of the MAPPO model to make it fit this change? And how to adjust the time step based on this change?
>
> Assume there are two agents, in the original MAPPO, the police of each agent is p(ai|s) and the value function is v(s). In our setting, the second level agent’s policy is p(a2|s, $ \emptyset $) and the value function is v(s, $ \emptyset $), we change the structure of the first level agent, its policy is p(a1|s, a2) and value function is v(s, a2). Other training procedures are the same. In practice, we use communication channels to achieve information sharing.
>
> We did not change the time step, since after all the agents make decisions, they execute the actions to interact with the environment simultaneously. Note that we did not break the fundamental dynamic, p(s’|s, a1,a2).
>
> >There is no code available in this submission, the details of the models' structures could not be seen. The reproducibility remains unsure.
>
> Many methods in communication-based MARL did not share the code, which makes the comparison very hard. We also do not like this trend, and we guarantee we will release the code as soon as this paper gets accepted. We also guarantee all the results can be reproduced.
>
> >Only comparing to TarMAC is not enough, this MARL with communication algorithm was proposed in 2019, and there are more recent MARL communication algorithms outperforming TarMAC.
>
> We add more baselines as requested in Figure 3 (pdf in the global response). Note that many methods in communication-based MARL did not share the code, meaning many baselines cannot be compared fairly. We choose two baselines for comparison (CommFormer [1] and MAIC[2]) with two criteria, 1. Published in the recent top conference; 2. Release the code for a fair comparison.
>
> The result shows that SeqComm still outperforms these baselines by a large margin. Surprisingly, we have tried our best to tuned the parameters under the limited timeline, but CommFormer (2024 baselines) still performs worsen than MAIC (2022 baselines).
>
> [1]. Learning Multi-Agent Communication from Graph Modeling Perspective. ICLR 2024
>
> [2]. Multi-Agent Incentive Communication via Decentralized Teammate Modeling. AAAI 2022

---

> > ### Comment · Reviewer_gF4i · 2024-08-09
> >
> > I thank the authors for addressing my concerns, I like the idea that the code will be released. I suggest that the authors should include the discussions about the new MDP in the main body of the paper.

---

> > > ### Author Response · Authors · 2024-08-11
> > >
> > > Thank you for your suggestion. We will include the discussions about the new MDP in the main body of the paper and release the code. If you find our clarifications satisfactory, we kindly ask for your consideration in adjusting your scores accordingly.

---

> > > > ### Comment · Reviewer_gF4i · 2024-08-14
> > > >
> > > > As stated in the authors' rebuttal, they formed a new MDP, and the formation of this new MDP necessitates the proof of the new alternative policy gradient theorem, which is currently absent in the manuscript. This proof is essential as it forms the foundational support for the proposed theorem.
> > > >
> > > > Additionally, the NeurIPS submission guidelines strongly recommend including code at the time of paper submission. The absence of accompanying code not only deviates from these guidelines but also undermines fairness, especially when compared to other submissions that included their codes. Therefore, I believe that my evaluation is justified and fair for this submission.

---

### Official Review · Reviewer_LzSK · 2024-07-12

**Soundness:** 3
**Presentation:** 3
**Contribution:** 3
**Rating:** 7
**Confidence:** 4

**Summary:**

This paper introduces SeqComm, a novel multi-level communication scheme for multi-agent coordination in reinforcement learning. The key contributions are:

1. A new approach treating agents asynchronously with two communication phases: negotiation and launching.
2. Theoretical analysis proving monotonic improvement and convergence of learned policies.
3. Empirical evaluation on StarCraft multi-agent challenge v2 (SMACv2) showing superior performance over existing methods.

The paper addresses the coordination problem in multi-agent settings by allowing agents to communicate actual actions and determine decision-making priorities dynamically.

**Strengths:**

The paper provides a solid theoretical foundation with proofs for policy improvement and convergence (Propositions 1 and 2, Theorem 1). The experimental methodology is sound, using the challenging SMACv2 benchmark and comparing against relevant baselines. The ablation studies support the importance of the proposed components.

The paper is generally well-written and organized. The introduction clearly motivates the problem and contributions. The method section provides a detailed explanation of SeqComm. Figures and algorithms help illustrate the approach.

The paper presents a novel and significant contribution to multi-agent reinforcement learning. The idea of asynchronous decision-making with dynamic priority determination is original and addresses an important challenge in the field. The theoretical guarantees and strong empirical results on a challenging benchmark demonstrate the value of the approach.

Overall the strengths are:
- Novel multi-level communication scheme addressing the coordination problem
- Theoretical analysis providing performance guarantees
- Fairly strong empirical results on SMACv2, outperforming existing methods
- Ablation studies demonstrating the importance of key components
- Addresses both full and local communication scenarios

I'd also like to point out that this was run on a GTX 1050, which makes the model approachable to a general audience.

**Weaknesses:**

- The reliance on homogeneous agents with parameter sharing may limit the applicability of SeqComm in real-world scenarios where agents often have diverse capabilities. However, it is a commonly accepted assumption.
- The approach heavily relies on a world model, but there's limited discussion on how the quality of this model affects performance, especially in more complex environments. Also the effect of the attention module is not analysed thoroughly. Attention mechanisms can be computationally expensive, especially as the number of agents increases. The paper doesn't address how this affects the overall scalability of SeqComm. It's not entirely clear how the attention mechanism is adapted for the local communication scenario, where agents only communicate with nearby agents. Furthermore, attention weights could potentially provide insights into which agents or information are most important for decision-making, but this aspect isn't explored in the paper.
- Overall, I believe a table reporting IQM values over all environments would strengthen the paper and help with reproducibility. Furthermore, plots showing wall-clock time in comparison to other models would be insightful, especially since final performance only appears to be significantly better in p_10v10 and t_10v10.
- While the paper addresses local communication, there's limited discussion on the trade-offs between communication frequency, accuracy, and overall system performance.
- Though not standard in related work either, the paper doesn't thoroughly explore the robustness of SeqComm to noisy observations, communication failures, or adversarial agents, which are common challenges in real-world multi-agent systems.
- While TarMAC is a strong baseline, the paper could benefit from comparisons with more recent state-of-the-art methods in multi-agent communication.

**Questions:**

1. How does the computational complexity of SeqComm scale with the number of agents, particularly in the negotiation phase? Is there a point where the overhead becomes prohibitive?
2. Have you explored the performance of SeqComm in heterogeneous agent scenarios where parameter sharing may not be applicable?
3. How might the method be adapted for such settings?
4. How sensitive is the approach to the quality of the world model, especially in more complex environments? What happens if the world model is significantly inaccurate?
5. The paper mentions that SeqComm can provide a proper priority of decision-making. How does this prioritization mechanism perform in highly dynamic environments where optimal decision order may change rapidly?
6. Have you investigated the potential emergence of behavioral patterns or strategies across agents due to the asynchronous decision-making process? Are there any interesting emergent behaviors?
7. How does the attention mechanism perform compared to simpler aggregation methods? An ablation study could illuminate this.
8. Does the attention mechanism learn meaningful patterns of inter-agent importance? Visualizing attention weights could provide insights into learned coordination strategies.
9. How does the computational cost of the attention mechanism scale with the number of agents, especially in the full communication scenario?
10. Could more advanced attention mechanisms (e.g., multi-head attention) provide further improvements?
11. How is the attention mechanism modified to handle the local communication scenario effectively?

**Limitations:**

- The evaluation is limited to one type of environment (SMACv2). It's unclear how well the method generalizes to other multi-agent scenarios with different dynamics or objectives.
- The heavy dependence on a world model could be a significant limitation in environments where accurate modeling is challenging or computationally expensive.
- While the paper addresses local communication, there's limited discussion on the trade-offs between communication frequency, accuracy, and overall system performance.
- The empirical results are not significantly better in a lot of environments and changing the sight range from SmacV2 makes comparisons to previous work harder.

---

> ### Author Rebuttal · Authors · 2024-08-07
>
> To begin with, we thank the reviewer for the carefully reviewing and insightful advice.
>
> >Computational complexity.
>
> The computational complexity of SeqComm is mainly related to communication overhead.
>
> For full communication, SeqComm needs more rounds, but it only transmits observation information for one time. For the rest n − 1 round communication with total (n − 1)/2 broadcasts per agent, only a single intention value and an action will be exchanged. Considering there are n! permutations of different order choices for n agents, our method has greatly reduced computation overhead since each agent needs to calculate up to n times to search for a satisfying order. Note that, in Section 3, we mentioned the cost-free communication setting. This extreme case gives us a better understanding of the benefit of communication, even if the results do not apply across all domains. Therefore, we propose a more applicable setting.
>
> For local communication, there are only two communication rounds. One is for sharing observations and information. Another is for intention values.
>
> >Heterogeneous agent scenarios.
>
> We would like to point out that the agents in SMACv2 can be heterogeneous. Each map has three types of agents. Three types have different functions (Ranged and melee types), and the types are randomly refreshed. It turns out our methods can be applied to heterogeneous agent scenarios. One policy can learn different strategies for different types of agents. Note that the observation contains the information of the agent type.
>
> >World model.
>
> Theorem 1 provides a useful relationship between the compounding errors and the police update. As long as we improve the return under the true dynamic by more than the gap (mentioned in line 292), we can guarantee the policy improvement under the world model. If no such policy exists to overcome the gap, it implies the model error is too high, that is, there is a large discrepancy between the world model and true dynamics. Thus, the order sequence obtained under the world model is not reliable. Such an order sequence is almost the same as a random one. Though a random order sequence also has the theoretical guarantee of Proposition 2, we have shown in Section 5.2 that a random order sequence leads to a poor local optimum empirically but it still can converge.
>
> >The prioritization mechanism
>
> In each step, we will recompute the order based on the current situation. The proposition 2 shows that, even the order changes each step, the monotonic improvement and convergence of the joint policy in SeqComm will not be affected.
>
> >Emergence of behavioral patterns
>
> We have visualized some key frames to illustrate the behavioral patterns in Figure 2 (pdf in the global response). In the combat game, the unified attack on one enemy is always more effective than dispersing attacks. From frames 1-3, agents have no target until one agent (at the end of the orange arrow) is close to an enemy (the bottom right corner). In frame 4, through the negotiation phase, that agent is chosen as the highest-level agent (level 5) since it has a better position to choose one enemy to attack. After lower-level agents obtained the actions of higher-level agents (illustrated by a white dashed line), all the red units ended their random roaming and instead launched a unified attack on the blue units.
>
>
> A similar behavioral pattern can also be observed in Frames 7-9.
>
> > Simpler aggregation methods.
>
> We have done the ablation study as requested in Figure 1 (pdf in the global response). It turns out the attention mechanism performs better than a simpler aggregation method. The reason is simple aggregation the observations will expand the dimension input of the neural network. It will impair the learning process since high-dimension input may contain many irrelevant information [1]. However, the attention mechanism helps focus on more important information by learning different weights on different observations.
>
> [1] Learning individually inferred communication for multi-agent cooperation. NIPS 2020.
>
> > Attention mechanism learns meaningful patterns.
>
> We have observed two patterns. One is upper-level agents will be highlighted since important actions are provided. Another is the agents that are far away will be overlooked.
>
> > How does the computational cost of the attention mechanism scale with the number of agents, especially in the full communication scenario?
>
> In full communication scenarios, the computation cost increases linearly with the number of agents.
>
> In the local communication scenarios, due to the limitation of the communication range, the number of local communicated agents will also be limited. Therefore, the communication cost has an upper bound.
>
> >More advanced attention mechanisms (e.g., multi-head attention).
>
> Yes, it may help to process the incoming messages from other agents. However, since the main contributions are not from the attention mechanism, we did not spend too much time investigating this. However, how the attention mechanisms or transformers architecture influences the learning process is another high-profile line of research in RL.
>
> > How is the attention mechanism modified to handle the local communication scenario effectively?
>
> The attention mechanism is originally used in natural language processing, where they are inherently capable of processing input of different lengths, just like recurrent neural networks.

---

> > ### Comment · Reviewer_LzSK · 2024-08-07
> >
> > I thank the authors for the extensive rebuttal and running additional results. Your replies certainly improved my outlook on the paper.
> >
> > One more weakness that I'd like your comment on is:
> > > Furthermore, plots showing wall-clock time in comparison to other models would be insightful, especially since final performance only appears to be significantly better in p_10v10 and t_10v10.
> >
> > I do not expect the wall-clock plots for each experiment, given that time is limited. However, I do believe it's an insightful comparison with baselines, as different communication techniques come with different computational costs. In the extreme, your method could take 100x the compute to achieve slightly better results. Alternatively, your method could be 100x faster, which would make the paper that much stronger. I agree that the ultimate metric is final training performance and wall-clock time is secondary, so this would not be a dealbreaker.

---

> > > ### Author Response · Authors · 2024-08-08
> > >
> > > Thank you for your feedback.
> > >
> > > We are sorry we missed this important question. Since the training speed between protoss, terran, and zerg are very small within the same method, we take the results in protoss maps to illustrate the difference.
> > >
> > > SeqComm (full comm) fps (frame per second during training):
> > >
> > > protoss 5v5: 64.32 10v10: 32.75 10v11: 29.82
> > >
> > > SeqComm (local comm) fps:
> > >
> > > protoss 5v5: 108.38 10v10: 64.03 10v11: 59.55
> > >
> > > MAPPO fps:
> > >
> > > protoss 5v5: 166\~167 10v10: 146\~147 10v11: 136\~137
> > >
> > > Note that all results are tested on NVIDIA A100.
> > >
> > > Compared with MAPPO, we only need 2x to 3x the compute to achieve better results.

---

> > > > ### Comment · Reviewer_LzSK · 2024-08-08
> > > >
> > > > Thank you for the quick turn-around. If the authors are willing to add a more extensive wall-clock comparison for the camera-ready version, I'd be happy to improve my score to a 7.
> > > >
> > > > A 7 expects a high impact on the community. I think this is justified as the authors provide convincing empirical and theoretical results that their proposed communication scheme is a valid research direction to pursue. The authors also claim to open-source the code upon acceptance, enabling reproducibility. Furthermore, the conceptual simplicity of the method in combination with the improved empirical results is usually a good indicator of impact. I think the 2x-3x wall-clock time is justifiable, especially with increasing computational efficiency of MARL environments. For example, SMAX provides a 40000x efficiency improvement over Smacv2 (33mins vs 44h) when training IPPO, which would make the 2x-3x slow down negligible [1]
> > > >
> > > > [1] Rutherford, Alexander, et al. "Jaxmarl: Multi-agent rl environments in jax." arXiv preprint arXiv:2311.10090 (2023).

---

> > > > > ### Author Response · Authors · 2024-08-08
> > > > >
> > > > > Thank you for the comment. We are willing to add a more extensive wall-clock comparison for the camera-ready version. We also guarantee that we will open-source the code upon acceptance, enabling reproducibility.

---

### Official Review · Reviewer_bJxi · 2024-07-12

**Soundness:** 3
**Presentation:** 4
**Contribution:** 3
**Rating:** 7
**Confidence:** 3

**Summary:**

This paper introduces SeqComm, a novel multi-level communication scheme for multi-agent reinforcement learning. SeqComm enables agents to coordinate asynchronously, with upper-level agents making decisions before lower-level ones. The approach involves two communication phases: negotiation and launching. In the negotiation phase, agents communicate hidden states to determine decision-making priority, while in the launching phase, upper-level agents lead in decision-making and share actions with lower-level agents.

**Strengths:**

1. SeqComm introduces a novel communication scheme that significantly improves coordination in multi-agent settings.
2. This paper demonstrates theoretical guarantees of monotonic improvement and convergence.
3. This method outperforms existing approaches in various cooperative tasks, showcasing its effectiveness in challenging environments.

**Weaknesses:**

1. The assumptions regarding local observations and communications might not be realistic for all applications.

**Questions:**

None

**Limitations:**

refer to the "weaknesses" part.

---

> ### Author Rebuttal · Authors · 2024-08-07
>
> >The assumptions regarding local observations and communications might not be realistic for all applications.
>
> Thank you for pointing out the limitations. We agree that the settings cannot be applied to all the applications. However, our setting is more realistic compared to other full communication and global observation settings.
>
> In more detail, local observations are widespread in the real world because of the limitations of distance, hardware, or other objective factors. For example, not all information can be measured and obtained by the sensors in the real world, which makes global observations unrealistic.
>
> For communication, it is widely used in the real world. People use wireless communication to access the Internet. Also, the Internet of Vehicles and the Internet of Things aim to connect all things with communication by 5G, which means many works believe communication is necessary despite the difficulty in implementation.

---

> > ### Comment · Reviewer_bJxi · 2024-08-12
> >
> > Thank you for your rebuttal that eliminates my concerns. I will increase my score to 7.

---

> > > ### Author Response · Authors · 2024-08-12
> > >
> > > Thank you for your approval, we will revise the text in conjunction with the rebuttal.

---

### Official Review · Reviewer_iQrt · 2024-07-26

**Soundness:** 3
**Presentation:** 3
**Contribution:** 3
**Rating:** 7
**Confidence:** 3

**Summary:**

This paper introduces SeqComm, a novel multi-agent reinforcement learning (MARL) method that addresses coordination issues through sequential decision-making and multi-level communication. The main contributions include:

1. A new asynchronous perspective on MARL, allowing agents to make decisions sequentially.
2. A two-phase communication scheme: negotiation phase for determining decision priorities, and launching phase for explicit coordination.
3. Theoretical guarantees of monotonic improvement and convergence for the learned policies.
4. Empirical evaluation on SMACv2 demonstrating superior performance over existing methods.

**Strengths:**

(1) Innovative methodology: The paper proposes a novel approach to MARL by introducing sequential decision-making and multi-level communication, which effectively addresses coordination issues.
(2) Theoretical foundation: The authors provide rigorous theoretical analysis, including proofs of monotonic improvement and convergence for the learned policies.
(3) Comprehensive empirical evaluation: The method is thoroughly evaluated on multiple maps in SMACv2, demonstrating consistent performance improvements over existing baselines.
(4) Ablation studies: The paper includes detailed ablation studies that validate the importance of dynamic decision priority determination.
(5) Practical considerations: The authors provide both a full communication version and a local communication version, addressing potential limitations in real-world applications.

**Weaknesses:**

(1) Computational complexity: The paper lacks a detailed analysis of the computational overhead introduced by the multi-level communication scheme, especially for large-scale multi-agent systems.
(2) Sensitivity analysis: There is no discussion on the sensitivity of the method to hyperparameters, such as the number of sampling trajectories (F) or the length of predicted future trajectories (H).
(3) Potential for deadlocks: The paper lacks a thorough discussion on the possibility of deadlocks in the proposed asynchronous mechanism, which is a critical consideration for any asynchronous system.

**Questions:**

(1) Could you elaborate on your choice of SMACv2 as the primary benchmark for evaluating SeqComm? Are there specific characteristics of SMACv2 that make it particularly suitable for demonstrating the advantages of your method? Additionally, do you believe the results from SMACv2 would generalize well to other MARL environments, and if so, why?
(2) Have you considered the possibility of deadlocks in your asynchronous mechanism? What measures, if any, have been implemented to prevent or resolve potential deadlocks?
(3) Can you provide an analysis of the computational complexity of SeqComm compared to existing methods, especially for large-scale multi-agent systems?
(4) How sensitive is SeqComm to the choice of hyperparameters, particularly F and H? Are there any guidelines for selecting these values?

**Limitations:**

The authors have made an effort to address some limitations of their work, which is commendable. They acknowledge that the assumption of access to other agents' local observations might not be realistic in all applications. This transparency is appreciated and aligns with the NeurIPS guidelines on discussing limitations.
However, there are several areas where the discussion of limitations could be expanded:

Scalability: The authors could further discuss how SeqComm's performance and computational requirements might change as the number of agents increases.
Generalizability: While SMACv2 is a valuable benchmark, the authors could address how well they expect their results to generalize to other MARL environments.
Potential for deadlocks: Given the asynchronous nature of the method, a discussion on the possibility of deadlocks and how they are prevented or resolved would be beneficial.
Hyperparameter sensitivity: The authors could elaborate on how sensitive their method is to the choice of hyperparameters, particularly F and H.

Regarding societal impact, the authors have not explicitly discussed potential negative consequences. While the immediate applications of this work may seem benign, it would be beneficial to consider and discuss potential misuse scenarios or unintended consequences of more efficient multi-agent coordination.

---

> ### Author Rebuttal · Authors · 2024-08-07
>
> To begin with, we thank the reviewer for the carefully reviewing and insightful advice.
>
> >Could you elaborate on your choice of SMACv2 as the primary benchmark for evaluating SeqComm?
>
> In cooperative MARL, SMAC is the most popular testbed for the centralized training and decentralized execution paradigm. Other important testbeds are Google Research Football (GRF) and Multiple Particle Environment (MPE).
>
> Comparing the SMAC and other testbeds (MPE and GRF), SMACv2 (upgraded version of SMAC) is the latest testbed and proposed in 2023. This work critiques the original benchmark for lacking stochasticity and meaningful partial observability (SMAC, MPE, and GRF) and claims they simplify the coordination. It deliberately increases the stochasticity, meaning partial observability, and conducts thorough experiments to verify it. Our method aims to address the coordination issue in MARL and this problem is mostly induced by the partial observability of the agents. Therefore, our method is possible to demonstrate the superiority on the benchmark requiring high-level coordination, otherwise, the performance gap is not obvious for the benchmark where agents do not even need to coordinate to finish the tasks.
>
> Compared with other benchmarks, SMACv2 is more challenging from the following points. 1. Complex observation dimension (hundreds). 2. Diversity (many maps and unit types with different functions). 3. Stochasticity (different designed start positions). 4. meaningful partial observability (mask many irrelevant information). We believe the results from SMACv2 would generalize well to other MARL environments suffering from coordination issues since we chose the most challenging benchmark in this field.
>
> >Have you considered the possibility of deadlocks in your asynchronous mechanism? What measures, if any, have been implemented to prevent or resolve potential deadlocks?
>
> Deadlock occurs when two or more agents obtain the same intention value at the same time, and then they need another rule to determine the priority. In our implementation, each agent is assigned an index, and the rule to break the deadlock is the agent with a small index to determine first.
>
> > Can you provide an analysis of the computational complexity of SeqComm compared to existing methods, especially for large-scale multi-agent systems?
>
> The computational complexity of SeqComm mainly related to communication overhead.
>
> For full communication, SeqComm needs more rounds, but it only transmits observation information for one time. For the rest n − 1 round communication with total (n − 1)/2 broadcasts per agent, only a single intention value and an action will be exchanged. Considering there are n! permutations of different order choices for n agents, our method has greatly reduced computation overhead since each agent needs to calculate up to n times to search for a satisfying order.
>
> For local communication, there are only two communication rounds. One is for sharing observations information. Another is for intention values.
>
> >How sensitive is SeqComm to the choice of hyperparameters, particularly F and H? Are there any guidelines for selecting these values?
>
> H is the length of the trajectory. Empirically, we found that 2-4 can lead to decent performance. For too long length, the error caused by the world model can be too large. Besides, since the order of each step can change, the observation after more steps is meaningless
>
> F is the number of the future trajectories. Theoretically, the higher the number of F, the more accurate the estimation, but the computational cost also increases. Therefore, in practice, we need to seek a trade-off.

---

> > ### Comment · Reviewer_iQrt · 2024-08-08
> >
> > Thank you for the authors' comprehensive rebuttal and additional results. The response has indeed improved my view of the paper. However, I still have some concerns regarding the Stackelberg Equilibrium (SE) as the foundational assumption for agent coordination in the proposed SeqComm scheme.
> >
> > Specific Points for Further Consideration:
> >
> > Rationale for SE in MARL: While the authors have addressed the application of SE in SeqComm, I would appreciate a more rigorous justification for choosing SE over other potential equilibrium concepts. It would be beneficial to explore how SE compares to alternative concepts that may be more naturally aligned with the decentralized and dynamic nature of MARL environments. A comparative analysis highlighting the specific advantages of SE in this context would strengthen the paper's theoretical foundation.
> > Assumptions of SE: The paper assumes that agents can achieve and maintain an SE, which may not always be realistic in scenarios where agents have limited or imperfect information about others' strategies. I encourage the authors to discuss the robustness of their approach when these assumptions are violated. Specifically: a) How does the SeqComm scheme perform under conditions of information asymmetry?
> > b) What mechanisms, if any, are in place to ensure the stability of the SE in dynamic MARL environments?
> > c) How does the approach handle scenarios where perfect information about other agents' strategies is not available?

---

> ### Author Response · Authors · 2024-08-11
>
> This paper aims to demonstrate SE is better than NE. In nature, under SE where agent makes decisions sequentially, parts of agents can obtain the extra information (actions of upper-level agents). Intuitively, if the agent has more valuable information, it will help make the decision.
>
> For other potential equilibrium, if it allows the agent to get extra valuable information and does not break the fundamental dynamic of the environment, it also can benefit the performance.
>
> Assumption issues.
> 1. If information asymmetry which means communication is not allowed, then the SeqComm is degraded to MAPPO.
>
>      If only parts of the information are asymmetrical, it will degrade to the local-communication version.
>
> 2. In our view, we will recompute the decision-making order based on the current situation during the execution phase. It will ensure stability.
>
> 3. When perfect information about other agents' strategies is not available, we can do the opponent modeling, in more detail, we can train a policy that predicts the actions of other agents. Since we can get the observations of others via communication, the opponent modeling can be precise.

---

> > ### Comment · Reviewer_iQrt · 2024-08-11
> >
> > Thank you for your comprehensive rebuttal. Your explanations have effectively addressed my concerns and substantially improved my assessment of the paper. As a result, I have decided to increase my score to 7.
> >
> > This decision is based on several key strengths of your work:
> >
> > Your SeqComm approach offers an innovative solution to multi-agent coordination through multi-level communication, firmly grounded in Stackelberg Equilibrium theory. The method's robust performance on SMACv2, coupled with a clear analysis of its communication efficiency, demonstrates its practical value in complex MARL scenarios.
> >
> > I am confident that your contribution has the potential to make a significant impact on the MARL field.

---

> > > ### Author Response · Authors · 2024-08-12
> > >
> > > Thank you for your approval, we will revise the text in conjunction with the rebuttal.

---

### Author Rebuttal · Authors · 2024-08-07

First of all, we are very grateful to the reviewers for their thorough review of our paper. We highly appreciate your valuable comments. We will emphasize the key points mentioned during the rebuttal period in the revised version.

Additionally, we have provided some extra experiments: Figure 1 includes the attention module vs. the aggregation method; Figure 2 shows some behavioral analysis; Figure 3 shows the comparison with additional baselines.

---

### Decision · Program_Chairs · 2024-09-25

**Decision:**

Accept (poster)

**Comment:**

This paper introduces a multi-level communication scheme named SeqComm for multi-agent reinforcement learning. SeqComm enables agents to coordinate asynchronously, with upper-level agents making decisions before lower-level ones. The approach involves two communication phases: negotiation and launching. In the negotiation phase, agents communicate hidden states to determine decision-making priority, while in the launching phase, upper-level agents lead in decision-making and share actions with lower-level agents. The paper provides a solid theoretical foundation with proofs for policy improvement and convergence. The paper is generally well-written and organized. The introduction clearly motivates the problem and contributions. The method section provides a detailed explanation of SeqComm. The approach offers an innovative solution to multi-agent coordination through multi-level communication, firmly grounded in Stackelberg Equilibrium theory. Regard to the contributions, I recommend to accept the paper.

In the rebuttal period, the authors claimed that “many methods in communication-based MARL did not share the code, meaning many baselines cannot be compared fairly”, but this statement was not true, e.g., the SARNet paper (Learning Multi-Agent Communication through Structured Attentive Reasoning, NeurIPS 2020) released their codes including lots of other baselines as well. Moreover, the statement is somehow contradicted to “we don't think code submission should be the core criterion for judging this paper”. If nobody makes their codes public, a lot of work would be (or even impossible) very difficult to be repeated. So, it would be much better that the authors could do what they said “We promise to publish the code”.